# Radiocarbon dating of alpine ice cores with the dissolved organic carbon (DOC) fraction

Ling Fang[1,2,3], Theo M. Jenk[1,3,*], Thomas Singer[1,2,3], Shugui Hou[4,5], Margit Schwikowski[1,2,3]

*Corresponding author: Theo M. Jenk (theo.jenk@psi.ch)

[1]Laboratory for Environmental Chemistry, Paul Scherrer Institute, CH-5232 Villigen PSI, Switzerland

[2]Department of Chemistry and Biochemistry, University of Bern, CH-3012 Bern, Switzerland

[3]Oeschger Centre for Climate Change Research, University of Bern, CH-3012 Bern, Switzerland

[4]School of Geographic and Oceanographic Sciences, Nanjing University, Nanjing, 210023, China

[5]School of Oceanography, Shanghai Jiao Tong University, Shanghai 200240, China

## Abstract

High-alpine glaciers are valuable archives of past climatic and environmental conditions. The interpretation of the preserved signal requires a precise chronology. Radiocarbon ($^{14}$C) dating of the water-insoluble organic carbon (WIOC) fraction has become an important dating tool to constrain the age of ice cores from mid-latitude and low-latitude glaciers. However, in some cases this method is restricted by the low WIOC concentration in the ice. In this work, we report first $^{14}$C dating results using the dissolved organic carbon (DOC) fraction, which is present at concentrations of at least a factor of two higher than the WIOC fraction. We evaluated this new approach by comparison to the established WIO$^{14}$C dating based on parallel ice core sample sections from four different Eurasian glaciers covering an age range of several hundred to around 20'000 years. $^{14}$C dating of the two fractions yielded comparable ages with WIO$^{14}$C revealing a slight, barely significant, systematic offset towards older ages comparable in magnitude with the analytical uncertainty. We attribute this offset to two effects of about equal size, but opposite in direction: (i) in-situ produced $^{14}$C contributing to the DOC resulting in a bias towards younger ages and (ii) incompletely removed carbonates from particulate mineral dust ($^{14}$C depleted) contributing to the WIOC fraction with a bias towards older ages. The estimated amount of in-situ produced $^{14}$C in the DOC fraction is smaller than the analytical uncertainty for most samples. Nevertheless, under extreme conditions, such as very high altitude and/or low snow accumulation rates, DO$^{14}$C dating results need to be interpreted cautiously. While during DOC extraction the removal of inorganic carbon is monitored for completeness, the removal for WIOC samples was so far only assumed to be quantitative, at least for ice samples containing average levels of mineral dust. Here we estimated an average removal efficiency of 98±2 %, resulting in a small offset in the order of the current analytical uncertainty. Future optimization of the removal procedure has the potential to improve the accuracy and precision of WIO$^{14}$C dating. With this study we demonstrate, that using the DOC fraction for $^{14}$C dating is not only a valuable alternative to the use of WIOC, but also benefits from a reduced required ice mass of typically ~250 g to achieve comparable precision of around ±200 years. This approach thus has the potential of pushing radiocarbon dating of ice forward even to remote regions where the carbon content in the ice is particularly low.

# 1 Introduction

For a meaningful interpretation of the recorded paleoclimate signals in ice cores from glacier archives, an accurate chronology is essential. Annual layer counting, supported and tied to independent time markers such as the 1963 nuclear fallout horizon evident by a peak maximum in tritium or other radioisotopes, or distinct signals from known volcanic eruptions in the past is the fundamental and most accurate technique used for ice core dating. However, for ice cores from high-alpine glaciers this approach is limited to a few centuries only, because of the exceptional strong thinning of annual layers in the vicinity of the bedrock. Most of the current analytical techniques do not allow high enough sampling resolution for resolving seasonal fluctuations or detecting distinct single events in this depth range. Ice flow models, which are widely used to retrieve full depth age scales (e.g. Nye, 1963; Bolzan, 1985; Thompson et al., 2006), also fail in the deepest part of high-alpine glaciers due to the assumption of steady state conditions and the complexity of glacial flow and bedrock geometry limiting realistic modeling of strain rates. Even with 3D models, which require extensive geometrical data, it is highly challenging to simulate a reasonable bottom age (e.g. Licciulli et al., 2020). This emphasizes the need for an absolute dating tool applicable to the oldest, bottom parts of cores from these sites.

Radioactive isotopes contained in the ice offer the opportunity to obtain absolute ages of an ice sample. For millennial scale ice cores, $^{14}$C dating is the technique of choice. With a half-life of 5370 years, dating in the age range from ~250 years to up to ten half-life times is theoretically possible, covering the time range accessible by alpine glaciers in the vast majority of cases (Uglietti et al., 2016). The $^{14}$C dating approach using water insoluble organic carbon (WIOC) from glacier ice has become a well-established technique for ice core dating and its accuracy was recently validated (Uglietti et al., 2016). Ice samples from mid- and low-latitude glaciers can now be dated with a reasonable uncertainty of 10-20%. Ice sample masses of 200-800 g are usually selected to aim for >10 μg carbon for $^{14}$C analysis with accelerator mass spectrometry (AMS), whereby the respective mass depends on sample age and organic carbon concentrations (Jenk et al., 2007; Jenk et al., 2009; Sigl et al., 2009; Uglietti et al., 2016; Hoffmann et al., 2018). Accordingly, the low WIOC concentration in some glaciers and in Polar Regions and the related large demands of ice mass puts a limit to this application. Concentrations of dissolved organic carbon (DOC) in glacier ice are a factor of 2-8 higher compared to typical WIOC concentrations (Legrand et al., 2007; Legrand et al., 2013; May et

al., 2013, Fang et al., in prep.). Using the DOC fraction for $^{14}C$ dating could therefore reduce
the required amount of ice or, for sample sizes similar to what would be needed for $^{14}C$ dating
by WIOC, improve the achievable analytical (dating) precision which strongly depends on the
absolute carbon mass even for state-of-the-art micro-radiocarbon dating.  The underlying
hypothesis of applying the DOC fraction for $^{14}C$ dating is the same as for the WIO$^{14}C$ dating
approach (Jenk et al., 2006; Jenk et al., 2007; Jenk et al., 2009). DOC in ice is composed of
atmospheric water soluble organic carbon (WSOC) contained in carbonaceous aerosol particles
and organic gases taken up during precipitation (Legrand et al., 2013).  WSOC is formed in the
atmosphere by oxidation of gases emitted from the biosphere or from anthropogenic sources
(Legrand et al., 2013; Fang et al. in prep.) and subsequent condensation of the less volatile
products. Carbonaceous aerosols transported in the atmosphere can be deposited on a glacier
by wet and dry deposition. Before the industrial revolution, these organic carbon species, then
entirely of non-fossil origin, contain the contemporary atmospheric $^{14}C$ signal of the time when
the snow deposited on the glacier (Jenk et al., 2006). For both WIOC and WSOC, carbon from
biomass burning and oceanic organic matter can potentially introduce a reservoir effect
(sources of aged carbon). The mixed age of trees in Swiss forests today is estimated to be
slightly less than 40 years (Mohn et al., 2008). Back in time, prior to extensive human forest
management, the mixed age of trees in Europe was likely older and the mean age of old-growth
forest wood ranged from around 70 to 300 years depending on the region, i.e. the tree species
present (Gavin, 2001, Zhang et al., 2017). Prior to the use of fossil fuels about 50% of WIOC
is estimated to originate from biomass burning (Minguillon et al., 2011). For biogenic DOC,
May et al. (2013) estimated a turnover-time of around 3 to 5 years, corresponding to a 20%
contribution from biomass burning. With a mean age of burned material (aged wood plus grass
and bushes) of 150±100 years, this results in a potential in-built age from biomass burning for
WIOC and DOC of 75±50 and 30±20 years, respectively. Such an in-built age is negligible
considering the analytical uncertainty, which is similarly the case for a bias from oceanic
sources, since concentrations of marine organic tracers are more than one order of magnitude
lower than terrestrial tracers for the vast majority of glacier sites. This conclusion is supported
by the fact that Uglietti et al. (2016) did not identify such a bias, when comparing WIO$^{14}C$
ages with ages derived by independent methods.

109        For analyzing DO$^{14}C$ in ice cores, one of the major limitations is the relatively low

extraction efficiency ranging from 64 % (Steier et al., 2013) to 96 % (May et al., 2013; Fang
et al., 2019) and the high risk of sample contamination (Legrand et al., 2013) potentially

introduced during drilling, storage, and sample processing. A first attempt to use DOC for $^{14}$C dating of ice samples was conducted by May (2009) using a set-up for a combined analysis of both, the DOC and WIOC fraction with subsequent radiocarbon micro-analysis. However, these first results suggested a potential in-situ production of $^{14}$C in the DOC fraction based on the obtained super modern F$^{14}$C values (i.e. F$^{14}$C values higher than ever observed in the recent or past ambient atmosphere). Building on these initial findings, May (2009) questioned the applicability of the DOC fraction for radiocarbon dating. Although the in-situ $^{14}$C production of $^{14}$CO and $^{14}$CO$_2$ in air bubbles contained in polar ice has been studied thoroughly and is well understood (Van de Wal et al., 1994; Lal et al., 1997; Smith et al., 2000), possible mechanisms of $^{14}$C in-situ production followed by formation of organic compounds are not and only few studies exist to date (Woon, 2002; Hoffmann, 2016). To further explore the potential of DO$^{14}$C for dating ice, a DOC extraction setup for radiocarbon analyses was designed and built at the Paul Scherrer Institut (PSI). In order to minimize potential contamination, the entire system is protected from ambient air by inert gas (helium) flow or vacuum. To maximize the oxidation efficiency, the PSI DOC methodology applies an ultraviolet (UV) photochemical oxidation step supported by addition of Fenton's reagent. The setup has been characterized by a high extraction efficiency of 96% and a low overall process blank being superior in the resulting blank to sample ratio compared to other systems (Fang et al., 2019). The system can handle samples with volumes of up to ~350 mL. With this volume, samples with DOC concentrations as low as 25-30 µg/kg can be analyzed, yielding the minimal carbon mass required for reliable $^{14}$C analysis (~10 µg C). Pooling samples from several subsequent extractions would be feasible, allowing dating of samples with lower DOC concentration. In this study, we evaluate $^{14}$C dating with the DOC fraction by comparing to results from the well- established and validated WIO$^{14}$C dating method. This is not only analytically highly challenging, but also because of the very limited availability of the precious sampling material needed in a rather large quantity (total for both fractions > 500 g), ideally covering a wide range of ages from a few hundred to several thousands of years. Here, we succeeded to analyze such parallel samples from four different Eurasian glaciers.

## 2 Sample preparation and $^{14}$C analysis

To validate the DOC $^{14}$C dating technique, a total of 17 ice sections from the deep parts of ice cores from the four glaciers Colle Gnifetti, Belukha, Chongce (Core 1), and Shu Le Nan Shan

(SLNS) were selected (Figure 1). They were sampled in parallel to directly compare DOC and
WIOC concentrations and [14]C dating results. The high-alpine glacier Colle Gnifetti is located
in the Monte Rosa massif of the Swiss Alps, close to the Italian border. A 76 m long core was
retrieved from the glacier saddle in September 2015 at an altitude of 4450 m asl.
(45°55'45.7''N, 7°52'30.5''E; Sigl et al., 2018), only 16 m away from the location of a
previously dated core obtained in 2003 (Jenk et al., 2009). The low annual net accumulation
rate at this site (~0.45 m w.e. yr$^{-1}$) provides access to old ice covering the Holocene (Jenk et
al., 2009). Four samples were selected from the bottom 4 m closest to bedrock (72-76 m depth).
The Belukha core was drilled in May/June 2018 from the saddle between the two summits of
Belukha (49°48'27.7''N, 86°34'46.5''E, 4055 m asl.), the highest mountain in the Altai mountain
range. The bedrock was reached and the total length of the core is 160 m. Three samples were
analyzed from the deepest part (158-160 m). Seven, and three samples were analyzed from the
deep parts of SLNS and Chongce, respectively. The SLNS ice core was retrieved in May 2010
from the south slope of the Shu Le Nan Shan Mountain (38°42'19.35''N, 97°15'59.70''E, 5337
m asl.). The bedrock was reached and the total length of the ice core is 81.05 m (Hou et al.,
submitted). The Chongce ice cap is located in the western Kunlun Mountains on the
northwestern Tibetan Plateau, covering an area of 163.06 km$^2$ with a volume of 38.16 km$^3$
(Hou et al., 2018). The ice analyzed in this study was sampled from Chongce Core 1, one of
three ice cores drilled in October 2012 (35°14'5.77''N, 81°7'15.34''E, 6010 m asl.). Two of those
cores reached bedrock with lengths of 133.8 m (Core 1) and 135.8 m (Core 2). In 2013, two
more ice cores were recovered from a higher altitude of 6100 m asl., reaching bedrock with
lengths of 216.6 m (Core 4) and 208.6 m (Core 5) (Hou et al., 2018). The annual net
accumulation rate is about 0.14 m w.e. yr$^{-1}$ for Core 3, located less than 2 km away from Core
1. A summary of the metadata for the study sites and ice cores can be found in the supplement
(Table S1) and details about sampling depths and sample sizes in Table 1. No results from any
of the cores analyzed in this study have been published previously.

171        All sampled ice sections were decontaminated in a cold room (-20ºC) by cutting off the

surface layer (~3 mm) and each section split into two parallel samples to perform both WIOC
and DOC [14]C analysis. Samples for WIO[14]C-dating were prepared following the protocol
described in Uglietti et al. (2016) with a brief summary provided in the following.  In order to
remove potential contamination in the outer layer of the ice core, pre-cut samples from the
inner part of the core were additionally rinsed with ultra-pure water (Sartorius, 18.2 MΩ×cm,
TOC < 5ppb), resulting in samples masses ranging from ~300 to 600 g (Table 1). To dissolve
carbonate potentially present in the ice, melted samples were acidified with HCl to pH < 2,
before being sonicated for 5 min. Subsequently, the contained particles were filtered onto pre-
baked (heated at 800 °C for 5 h) quartz fiber filters (Pallflex Tissueqtz-2500QAT-UP). In a
second carbonate removal step, the filters were acidified 3 times with a total amount of 50 μL
0.2M HCl, left for 1 h, rinsed with 5 mL ultra-pure water and finally left again for drying.
These initial steps were performed in a laminar flow box to ensure clean conditions. At the
Laboratory for the Analysis of Radiocarbon with AMS (LARA) of the University of Bern the
particle samples were then combusted in a thermo-optical OC/EC analyzer (Model4L, Sunset
Laboratory Inc, USA) equipped with a non-dispersive infrared (NDIR) cell to quantify the $CO_2$
produced, using the well-established Swiss 4S protocol for OC/EC separation (Zhang et al.,
2012). Being coupled to a 200 kV compact accelerator mass spectrometer (AMS, MIni CArbon
DAting System MICADAS) equipped with a gas ion source via a Gas Interface System (GIS,
Ruff et al., 2007; Synal et al., 2007, Szidat et al., 2014), the LARA Sunset-GIS-AMS system
(Agrios et al., 2015; Agrios et al., 2017) allowed for final, direct online [14]C measurements of
the $CO_2$ produced from the WIOC fraction.
For DO[14]C analysis, sample preparation follows the procedure described in Fang et al.
(2019). After transfer of pre-cut samples to the laboratory and before being melted, samples
were further decontaminated in the pre-cleaned melting vessel of the extraction setup by rinsing
with ultrapure water (sample mass loss of about 20-30 %), all performed under helium
atmosphere. Simultaneously, a pre-cleaning step was applied to remove potential
contamination in the system. For this, 50 mL ultra-pure water was injected into the reactor and
acidified with 1 mL of 85% $H_3PO_4$. To enhance the oxidation efficiency, 2 mL of 100 ppm
$FeSO_4$ and 1 mL of 50 mM $H_2O_2$ (Fenton's reagent) was also injected into the base water
before turning on the UV lights for ~20 min, thereby monitoring the process via the online
NDIR $CO_2$ analyzer. After the ice melted, the meltwater was filtrated under helium atmosphere,
using a pre-baked in-line quartz fiber filter. The sample volume was determined by measuring
the reactor fill level. The filtrate was acidified by mixing with the pre-treated base water. After
degassing of $CO_2$ from inorganic carbon was completed as monitored by the $CO_2$- detector, 1
mL of 50 mM $H_2O_2$ was injected into the reactor right before the irradiation started. During
UV oxidation, water vapor was removed by cryogenic trapping at -60 °C and produced $CO_2$
was trapped in liquid nitrogen. All steps were carried out under a constant flow of helium. The
sample $CO_2$ was further cleaned from residual water vapor and quantified manometrically
before being sealed into a glass vial for offline [14]C analyses. The $CO_2$ gas from DOC in the

glass vial was directly injected into the MICADAS using a cracking system for glass vials under vacuum, allowing to then carry the $CO_2$ gas in a helium flow to the AMS ion source (Wacker et al., 2013). Procedural blanks were determined and continuously monitored by processing and analyzing frozen ultra-pure water (Sartorius, 18.2 MΩ cm, TOC< 5ppb) similar to natural ice samples. They were prepared every time when cutting ice and then processed/analyzed along with the samples at least twice a week. Procedural blanks are 1.3±0.6 µg C with an $F^{14}C$ of 0.69±0.15 (n=76) and 1.9±1.6 µg C with an $F^{14}C$ value of 0.68±0.13 (n=30) for WIOC and DOC, respectively.

All $^{14}C$ results are expressed as fraction modern ($F^{14}C$), which is the $^{14}C/^{12}C$ ratio of the sample divided by the same ratio of the modern standard referenced to the year 1950 (NIST, SRM 4990C, oxalic acid II), both being normalized to -25‰ in $\delta^{13}C$ to account for isotopic fractionation. All AMS $F^{14}C$ values presented here are finally corrected for the system and method characteristic contributions as reported previously (e.g. Uglietti et al., 2016 and Fang et al., 2019). For WIOC analysis using the Sunset-GIS-AMS system this includes a correction for the system background, i.e. constant contamination (0.91±0.18 µgC with $F^{14}C$ of 0.72±0.11). For the cracking system applied for DOC samples the constant contamination is 0.06±0.18 µgC with $F^{14}C$ of 0.50 ±0.11). Further corrections applied account for the AMS cross contamination (0.2% of the previous sample), and procedural blanks (see above). All uncertainties were propagated throughout data processing until final $^{14}C$ calibration. These corrections, have a larger effect on low carbon mass samples (higher noise-to-sample ratio), resulting in a larger dating uncertainty. Therefore, we only discuss samples with a carbon mass larger than 10 µg as recommended in Uglietti et al. (2016). Radiocarbon ages are calculated following the law of radioactive decay using 5570 years as the half-life of radiocarbon, thus age equals -8033 * ln ($F^{14}C$) with -8033 years being Libby's mean lifetime of radiocarbon. Radiocarbon ages are given in years before present (BP) with the year of reference being 1950 (Stuiver and Polach, 1977). To obtain calibrated $^{14}C$ ages, the online program OxCal v4.3.2 with the IntCal13 radiocarbon calibration curve was used (Reimer et al., 2013; Ramsey, 2017). Calibrated ages, also given in years before present, are indicated with (cal BP) and denote the 1σ range unless stated otherwise.

## 3 Results

## 3.1 DOC and WIOC concentrations

DOC concentrations are generally higher compared to the corresponding WIOC concentrations (Figure 2). For all samples from the four glaciers, the DOC/WIOC concentration ratio ranges from 1.2 to 4.0 with an average of 1.9 ± 0.6 (Table 2). This is at the lower end of previously reported average DOC/WIOC ratios of 2-8 (Legrand et al., 2007; Legrand et al., 2013, Fang et al., in prep.). This is likely explained by temporal variability because most samples in this study are several thousand years old, whereas the literature data only covers the last few centuries, including values from the industrial period in which additional anthropogenic sources exist (e.g. fossil DOC precursors). It is interesting to note that the average DOC/WIOC ratio at Belukha (2.5) is higher compared to the other sites (Colle Gnifetti, SLNS and Chongce is 1.8, 1.7 and 1.6, respectively). Because the Belukha glacier is surround by extensive Siberian Forests, the higher ratio may be explained by particularly high emissions of biogenic volatile organic compounds. This is corroborated by the observation that DOC concentrations are highest at this site (241 ± 82 µg/kg) (Figure 2). Absolute concentrations of DOC and WIOC are slightly lower at Colle Gnifetti (112 ± 12 µg/kg and 63 ± 13 µg/kg, respectively) compared to the other three glaciers (Table 1 and 2). Mean DOC and WIOC concentrations in the ice from the Tibetan Plateau are 211±28 µg/kg and 123±19 µg/kg for SLNS and 156±40 µg/kg and 99±37 µg/kg for Chongce, respectively. These values are higher compared to the pre-industrial (PI) average values found in European Alpine glaciers, not only compared to the few samples from Colle Gnifetti of this study, but also to previously reported values from the Fiescherhorn glacier with PI-DOC of ~95 µg/kg (Fang et al., in prep.) and PI-WIOC of ~30 µg/kg (Jenk et al., 2006), respectively; and from Colle Gnifetti with PI-WIOC of ~30 µg/kg (Legrand et al., 2007; Jenk et al., 2006).

## 3.2 Radiocarbon results

For all four sites, $F^{14}C$ of both fractions (WIOC and DOC) decreases with depth, indicating the expected increase in age (Figure 2, Table 1 and 2). For three of the sites (Colle Gnifetti, Belukha and SLNS), the corresponding DOC and WIOC fractions yielded comparable $F^{14}C$ values with no statistical evidence for a significant difference (Mann-Whitney U-test, U=79.5, n=14, p=0.41>0.05). They scatter along the 1:1 ratio line, are significantly correlated (Pearson correlation coefficient r=0.986, p < .01, n=14) and both intercept (0.025 ± 0.034) and slope

(1.034 ± 0.050) are not significantly different from 0 and 1, respectively (Figure 3).
Nevertheless, a slight systematic offset towards lower $F^{14}C$ values for WIOC compared to
DOC seems evident if looking at Figures 2 and 3. This is particularly obvious for the samples
from Chongce, characterized by high mineral dust load and from a site of very high elevation
with low net accumulation. For these samples, the $F^{14}C$ DOC-WIOC offset is significant
(discussion in Sect. 4.2 and 4.3).
For all sites, the calibrated $^{14}C$ ages from both fractions show an increase in age with
depth (Table 3). The ages range from ~0.2 to 20.3 kyr cal BP for DOC and ~0.8 to 22.4 kyr cal
BP for WIOC, respectively. In both fractions, the oldest age was derived for the sample from
the deepest part of the Belukha ice core. Samples from Colle Gnifetti generally showed younger
ages (< 2 kyr cal BP). The two ice cores from the Tibetan Plateau (SLNS and Chongce) cover
a similar age span from ~0.2±0.1 to 5.5±0.3 kyr cal BP in the DOC fraction. WIO$^{14}C$ resulted
in a similar age range for the samples from SLNS (0.8±0.4 to 6.6±0.8 kyr cal BP), but was
considerably older for Chongce (3.1±0.7 to 11.0±1.7 kyr cal BP, discussion in Sect. 4.2 and
290  4.3).


**4 Discussion**

**4.1 Radiocarbon dating with the DOC fraction**
In Table 3, we present the first radiocarbon dating results of ice using the DOC fraction. The
DOC calibrated $^{14}C$ age of ice increases with depth for all four sites, as expected for
undisturbed glacier archives from the accumulation zone. For samples from three out of the
four sites, our results (Sect. 3) indicate no significant difference in $F^{14}C$ between DOC and
WIOC, with the latter fraction being validated for allowing accurate dating of the surrounding
ice (Uglietti et al., 2016). With the new DO$^{14}C$ dating method an average dating uncertainty of
around ±200 years was achieved for samples with an absolute carbon mass of 20-60 μg and ice
younger than ~6 kyrs (Table 2 and 3). The analytical uncertainty mainly arises from correction
for the procedure blank introduced during sample treatment prior to AMS analysis (see Sect. 2
for details about other corrections), contributing with 20 to 70 % to the final overall dating
uncertainty. The contribution thereby depends on carbon mass (larger for small samples) and
sample age (larger the bigger the difference between sample and blank $F^{14}C$). How the overall
analytical uncertainty of $F^{14}C$ decreases with higher carbon mass is shown in Figure S1. For

DOC concentrations observed in this study, an initial ice mass of about 250 g was required, with about 20-30 % of the ice being removed during the decontamination processes inside the DOC set-up, yielding ~200 g of ice available for final analysis. Expected based on previously reported DOC/WIOC concentration ratios (Sect. 3.1), the results here confirmed that with this new technique, the required ice mass can be reduced by more than a factor of two compared to the mass needed for $^{14}$C dating using the WIOC fraction. Consequently, using the DOC instead of the WIOC fraction for $^{14}$C dating, a higher dating precision can be achieved for ice samples of similar mass. An additional benefit is that the DOC extraction procedure allows monitoring the removal of inorganic carbon for completeness (see Sect. 2), which is important to avoid a potential age bias (see Sect. 4.3).

## 4.2 Potential contribution of $^{14}$C in-situ production to DO$^{14}$C

Previous studies suggested that $^{14}$C of the DOC fraction may be influenced by in-situ production of $^{14}$C in the ice matrix (May 2009; Hoffman 2016). Induced by cosmic radiation, the production of $^{14}$C atoms within the ice matrix, i.e. by spallation of oxygen within the water molecule, is a well-known process (Lal et al., 1987; Van de Wal et al., 1994). Earlier studies indicated that in-situ produced $^{14}$C atoms mostly form CO, $CO_2$ and $CH_4$ (Petrenko et al., 2013), but also can form methanol and formic acid (Yankwich et al., 1946, Woon, 2002). The mechanism of incorporation of in-situ produced $^{14}$C incorporation into organic molecules is not well understood (Woon, 2002; Hoffman, 2016). Hoffmann (2016) performed neutron irradiation experiments on Alpine glacier ice, showing that about 11-25 % of the initially produced $^{14}$C atoms entered into the DOC fraction. The resulting effect on F$^{14}$C of DOC consequently depends on (i) the number of $^{14}$C atoms produced in the ice ($^{14}$C in-situ production), (ii) the fraction of these atoms incorporated into DOC, and because F$^{14}$C is based on a $^{14}$C/$^{12}$C ratio, (iii) the DOC concentration in the ice (the higher the smaller the resulting shift in F$^{14}$C-DOC).

The natural neutron flux, relevant for the $^{14}$C production rate, strongly depends on altitude and latitude with a generally uniform energy distribution of the incoming neutrons (Gordon et al., 2004). The $^{14}$C in-situ production in natural ice further depends on the depth in the glacier and the snow accumulation rate of the site (Lal et al., 1987), determining the totally received neutron radiation. Following Lal et al. (1987), the number of in-situ produced $^{14}$C

atoms in each of our ice samples was estimated, assuming an average incorporation into DOC of 18±7 % (Hoffmann, 2016) (Table 4, equations and input parameters in the Supplementary Material). The average $F^{14}C$-DOC shift for all samples is 0.044±0.033. We find a good correlation between the measured $F^{14}C$ DOC-WIOC offset and the $^{14}C$ in-situ caused $F^{14}C$-DOC shift which explains about 50 % of the offset (Pearson r=0.82, Figure 4) and after correcting for it improves the overall agreement between $F^{14}C$ of DOC and WIOC (Figure 5). The shift is largest for the Chongce samples (0.109±0.048) as a result of the high production rate at 6 km altitude in combination with the low annual net accumulation rate at this site (0.14 m w.e. yr$^{-1}$). The calculated shift for samples from the SLNS core, from similar latitude but from a site lower in altitude (5 km) and experiencing higher net accumulation (0.21 m w.e. yr$^{-1}$), is significant lower with 0.038±0.016. The samples from Belukha and Colle Gnifetti are least affected (0.013±0.006 and 0.033±0.013, respectively).

We find that while the effect of in-situ $^{14}C$ production causes only a negligible shift in $F^{14}C$-DOC for most samples (masked by the analytical uncertainty), it can become significant for ice samples from sites of exceptional high altitude and experiencing low annual net accumulation rates in addition, such as the Chongce ice cap (6010 m asl., 0.14 m w.e. yr$^{-1}$; Figure 4). Note that for any site, the size of this effect gets reduced the higher the DOC concentration of the sample.

## 4.3 Potential contribution of carbonates to $^{14}C$ of WIOC

Under the basic assumption that the initially emitted fractions of DOC and WIOC are of similar age, an additional contribution from $^{14}C$-depleted carbonate (low $F^{14}C$) to the WIOC would cause an $F^{14}C$ offset between the two fractions. Previously published WIOC $^{14}C$ ages from the upper parts of the Chongce Core 2 and Core 4, less than 2 and ~6 km away from Core 1, did show large scatter with no clear increase in age with depth for samples younger than 2 kyrs. It was speculated that this was at least partly caused by the visible, exceptionally high loading of mineral dust on the WIOC filters (Hou et al., 2018). Such high mineral dust loading was also observed during filtration of the Chongce Core 1 samples presented here. High mineral dust content in the ice can influence $^{14}C$ dating with WIOC in two ways, by affecting filtration through clogging of the filter and by potentially contributing with $^{14}C$-depleted carbon from carbonate, as has been discussed in most previous studies. They all concluded, that although

for dust levels typically observed in ice cores from high elevation glaciers no significant bias
is detectable for $^{14}$C of WIOC, it is of concern for the elemental carbon (EC) fraction
combusted at higher temperatures during OC/EC separation. EC – as well as total carbon (TC,
the sum of OC and EC) – is thus not recommended to be used for radiocarbon dating (Jenk et
al., 2006; Jenk et al., 2007; Jenk et al., 2009; Sigl et al., 2009; Uglietti et al., 2016). In any case,
the carbonate removal efficiency during WIOC sample preparation was never quantified.

378        Here, the hypothesis that incomplete removal of carbonate may have caused the F$^{14}$C

DOC-WIOC offset remaining after accounting for DO$^{14}$C in-situ production (Sect. 4.2) was
tested. Applying an isotopic mass balance based model to our dataset, the carbonate removal
efficiency in WIO$^{14}$C samples was estimated. The $Ca^{2+}$ concentration in the ice samples was
thereby used as a tracer for calcium carbonate (see Supplement for details).

383        We find a carbonate removal procedure incomplete by around 2 % (i.e. an average

removal efficiency of 98±2 %) to be sufficient for explaining the remaining part of the observed
F$^{14}$C DOC-WIOC offset Figure 5). In terms of residual carbonate carbon mass on the filter,
this equals to < 2 μgC on average (Table S2). On the one hand, this is in agreement with the
findings of previous studies, indicating that the potential carbonate related bias for $^{14}$C dating
using WIOC is hardly detectable for ice samples with normal dust loading (effect masked by
the analytical uncertainty, see Figure S2). For example, Uglietti et al. (2016) did not detect
such an effect when successfully validating WIO$^{14}$C dating results with ages from independent
methods. On the other hand, it demonstrates that a slightly below average removal efficiency
for ice samples containing visibly high loading of mineral dust can already cause a notable
offset (93-97 % for Chongce). The likely bigger particle size in such samples will affect their
solubility, i.e. increase the dissolution time required in the acid treatment step. In the current
procedure, this time is not adjusted accordingly (Sect. 2). Based on these results, we consider
a small offset from incomplete carbonate removal to be a very likely reason contributing to the
measured F$^{14}$C DOC-WIOC, i.e. resulting dating offset (Figure 5). Instead of a correction,
which does not seem feasible for this effect because of large uncertainties and likely substantial
site-to-site (sample-to-sample) variations, we suggest future improvement in the analytical
procedure of the carbonate removal step (e.g. a slight increase in acid concentration and an
increase of the reaction time).

**4.4 DO$^{14}$C ages in the context of published chronologies**

In the following we will discuss our new DO$^{14}$C results in the context of ages from previous studies. For final calibration of $^{14}$C ages, most of those earlier studies took advantage of the assumption of sequential deposition in the archive, i.e. a continuous, undisturbed and preserved sequential deposition of annual snow layers on the glacier surface. Particularly in case of relatively large analytical uncertainties compared to the age difference of the samples, the sequential deposition model can moderately constrain the probability distribution of the calibrated age range in each sample of the dataset. For consistency we applied the same calibration approach here by using the in-built OxCal sequence model (Ramsey, 2008). While the underlying assumption may not generally be valid for all sites, and individually needs to be carefully assessed, we find no difference in the calibrated ages using the sequence model and the ages from the conventional calibration approach for all DO$^{14}$C data presented in this study (Table 3). Note, that no correction for a potential in-situ $^{14}$C bias was applied to the DO$^{14}$C data used here (Section 4.2).

We obtained the oldest age of ~21 kyr cal BP for the bedrock ice at Belukha, indicating this glacier to be oldest and of Pleistocene origin. This is older than the previously reported age of ~11 kyr cal BP (Table 5, Figure 6). The latter age was obtained for an ice core from the nearby Belukha West Plateau glacier extracted in 2003 (B03) (Aizen et al., 2016; Uglietti et al., 2016) opposed to the 2018 core extracted from the saddle (B18) analyzed in this study. Also, the according sample from B03 was from a slightly shallower depth (0.6-0.3 m above bedrock) than the sample analyzed from B18 in this study. The age range modeled for B03 for the same depth above bedrock (0.5-0 m) is in better agreement with ~28 kyr cal BP and a very large uncertainty of ~15 kyr (Uglietti et al., 2016). Overall, our new age for the oldest ice at Belukha thus reasonably agrees with the previous result but yields a much better constrained age with a reduced uncertainty of ±4 kyr. The two glaciers from the Tibetan Plateau (SLNS and Chongce) show very similar bottom ages of ~5-6 kyr cal BP (Figure 6), which is in agreement with the previously reported age range of Tibetan Plateau glaciers (Hou et al., 2018). The bottom age of Chongce Core 1 determined here based on DO$^{14}$C (5.6 ± 0.3 kyr cal BP) is slightly younger than the previously reported bottom age in Core 2 based on WIO$^{14}$C (6.3 ± 0.3 kyr cal BP, Hou et al., 2018), which is in agreement with the findings discussed in Section 4.2 and 4.3. Nevertheless, our new age is still in the range of the previously estimated bottom age (Table 5, Figure 6). The bottom most sample of the Colle Gnifetti 2015 (CG15) core could not be dated because the small amount of ice available yielded an insufficient carbon mass of <10 μg for $^{14}$C analysis. Previous WIO$^{14}$C dating of a core obtained at Colle Gnifetti in 2003

(CG03) also revealed ice of Pleistocene origin with the ice at bedrock being older than 15 kyr
cal BP (Jenk et al., 2009). As expected, the age obtained in this study from a shallower depth
was much younger with 1.2 kyr cal BP. This is in excellent agreement with the age of CG03
for a similar depth (~74 m below surface; Table 5, Figure 6). We consider this as a clear
indication that the CG15 ice core did not reach bedrock.
Overall, the dating with $DO^{14}C$ results in ages which are in good agreement with the
age ranges reported in earlier studies. Even though a contribution from in-situ $^{14}C$ to $DO^{14}C$
was not considered in the comparison here, we find that the dating by the DOC fraction does
not lead to significantly different results compared to dating by $WIO^{14}C$ or cause a different
interpretation about the oldest ice still present for any of the sites.

## 5 Conclusion

In this study, we evaluated and successfully validated the $DO^{14}C$ dating technique by direct
comparison of dating results with the well-established $WIO^{14}C$ method using parallel ice
samples. Achieving this goal was not only analytically demanding but also highly challenging
due to the very limited availability of the sampling material, requiring ice in rather large
quantities and spanning a wide range of ages. The obtained $DO^{14}C$ ages for four different
Eurasian glaciers, ranging from 0.2 ± 0.2 to 20.3 ± 4.1 kyrs cal BP, agreed well with the
respective $WIO^{14}C$ ages (0.8 ± 0.4 to 22.4 ± 1.1 kyrs cal BP) and with previously published
chronologies from these ice core sites. This underlines the great potential for applying $DO^{14}C$
analysis for ice core dating. With this new method, an average dating uncertainty of around
±200 years was achieved for samples with an absolute carbon mass of > 20 μg and ages up to
~6 kyrs. For DOC concentrations observed in this study, an initial ice mass of about 250 g was
required. Our data confirmed previous results that concentrations of pre-industrial DOC are
higher by about a factor two compared to WIOC concentrations in high alpine ice cores. This
shows that the required ice mass to achieve similar precision is reduced by at least a factor of
two for $^{14}C$ dating when using the DOC instead of the WIOC fraction. Accordingly, an
improvement in precision can be achieved for same sample mass. Compared to WIOC, a
downside of using the DOC fraction for $^{14}C$ dating is a more demanding and time consuming
extraction procedure. In addition, because of its higher solubility and a related higher mobility
of DOC in case of meltwater formation, this fraction is only applicable for dating ice which

had been cold throughout its "lifetime". Beneficial compared to WIOC, there is no potential for a dating bias by carbonates of mineral dust for $DO^{14}C$. However, our results confirm previously suggested potential dating biases from in-situ $^{14}C$ causing $DO^{14}C$ dates to shift towards younger ages. While we find the effect to be small (at the level of analytical uncertainty), it may become significant for $DO^{14}C$ dating of ice samples from sites of e.g. exceptional high altitude, experiencing low annual net accumulation rates in addition. For such sites, a reasonably accurate correction to account for the age bias seems feasible according to our results, although at the cost of an increase in the final dating uncertainty. Nevertheless, we think this new dating method has a great potential to open up new fields for radiocarbon dating of ice for example from remote regions, where concentrations of organic impurities in the ice are particularly low.

**Acknowledgements**

We thank Johannes Schindler for his great work in designing and building the DOC extraction system and the two drilling teams on Colle Gnifetti and Belukha for collecting high quality ice cores. We acknowledge funding from the Swiss National Science Foundation (SNF) for the Sinergia project Paleo fires from high-alpine ice cores (CRSII2_154450), which allowed ice core drilling on Colle Gnifetti and Belukha, for the project Radiocarbon dating of glacier ice (200021_126515), and for the project Reconstruction of pre-industrial to industrial changes of organic aerosols from glacier ice cores (200021_182765). We acknowledge the funding from the National Natural Science Foundation of China (91837102, 41830644) for the Tibetan ice core drilling. We dedicate this study to Alexander Zapf, who died tragically while climbing in the Swiss Alps, before he could fulfil his dream of ice dating with $DO^{14}C$.

**Data availability**

The data is provided in the Tables.

**Author contributions**

LF and TS performed $^{14}C$ analysis. LF, TS, TMJ, and MS wrote the manuscript while all authors contributed to the discussion of the results. MS designed the study.

**Competing interests**

499     The authors declare that they have no conflict of interest.

500

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

**Table 1** WIOC samples analyzed from Colle Gnifetti, Belukha, SLNS and Chongce ice cores.

| Core section | Depth (m) | Ice mass (kg) | WIOC (µg) | Concentration (µg/kg) | Bern AMS Nr. | $F^{14}C$ (±1σ) | $^{14}C$ age (BP, ±1σ) |
|---|---|---|---|---|---|---|---|
| CG110 | 72.1-72.7 | 0.570 | 35.2 | 61.9±3.3 | 11770.1.1 | 0.875±0.011 | 1073±105 |
| CG111 | 72.7-73.4 | 0.539 | 38.7 | 71.8±3.8 | 11771.1.1 | 0.848±0.011 | 1321±101 |
| CG112 | 73.4-73.9 | 0.536 | 23.7 | 44.1±2.4 | 11772.1.1 | 0.852±0.015 | 1284±143 |
| CG113 | 73.9-74.6 | 0.549 | 39.8 | 72.4±3.8 | 11773.1.1 | 0.786±0.011 | 1937±109 |
| Belukha412 | 158.3-159.0 | 0.443 | 37.8 | 85.2±4.5 | 11766.1.1 | 0.367±0.010 | 8055±211 |
| Belukha414 | 159.5-160.3 | 0.336 | 27.8 | 82.6±4.4 | 11768.1.1 | 0.212±0.014 | 12473±535 |
| Belukha415 | 160.3-160.9 | 0.319 | 39.3 | 123.3±6.5 | 11769.1.1 | 0.100±0.011 | 18462±899 |
| SLNS101 | 56.8-57.5 | 0.420 | 41.5 | 98.9±2.1 | 12325.1.1 | 0.902±0.047 | 825±420 |
| SLNS113 | 64.7-65.4 | 0.427 | 45.3 | 106.1±2.5 | 12324.1.1 | 0.852±0.046 | 1284±438 |
| SLNS122 | 68.9-69.7 | 0.424 | 58.5 | 138.0±3.6 | 12323.1.1 | 0.807±0.046 | 1727±459 |
| SLNS127 | 71.8-72.5 | 0.483 | 50.9 | 105.3±2.5 | 12322.1.1 | 0.695±0.046 | 2921±532 |
| SLNS136 | 76.7-77.5 | 0.374 | 50.6 | 135.2±3.0 | 12321.1.1 | 0.521±0.046 | 5235±706 |
| SLNS139 | 78.9-79.6 | 0.485 | 61.2 | 126.3±3.6 | 12320.1.1 | 0.521±0.045 | 5232±703 |
| SLNS141-142 | 80.3-81.0 | 0.413 | 61.7 | 149.5±3.8 | 12319.1.1 | 0.489±0.046 | 5754±750 |
| CC237 | 126.0-126.7 | 0.352 | 22.4 | 63.7±1.8 | 12328.1.1 | 0.704±0.049 | 2815±555 |
| CC244 | 130.2-130.8 | 0.311 | 29.8 | 95.9±2.2 | 12327.1.1 | 0.639±0.048 | 3602±600 |
| CC252 | 133.4-133.8 | 0.174 | 23.8 | 136.7±4.3 | 12326.1.1 | 0.316±0.049 | 9256±1250 |

**Table 2** DOC samples analyzed for Colle Gnifetti, Belukha, SLNS and Chongce ice cores.

| Core section | Depth (m) | Ice mass (kg) | DOC (µg) | Concentration (µg/kg) | Bern AMS Nr. | F$^{14}$C (±1σ) | $^{14}$C age (BP, ±1σ) | DOC/WIOC |
|---|---|---|---|---|---|---|---|---|
| CG110 | 72.1-72.7 | 0.171 | 18.9 | 110.0±2.7 | 11575.1.1 | 0.943±0.030 | 474±259 | 1.8 |
| CG111 | 72.7-73.4 | 0.207 | 25.5 | 122.9±3.0 | 11576.1.1 | 0.901±0.021 | 836±190 | 1.7 |
| CG112 | 73.4-73.9 | 0.248 | 23.6 | 95.0±2.3 | 11577.1.1 | 0.889±0.021 | 943±192 | 2.2 |
| CG113 | 73.9-74.6 | 0.246 | 29.5 | 119.4±2.9 | 11578.1.1 | 0.849±0.016 | 1312±151 | 1.7 |
| Belukha412 | 158.3-159.0 | 0.172 | 28.5 | 165.0±4.0 | 11581.1.1 | 0.315±0.024 | 9284±624 | 1.9 |
| Belukha414 | 159.5-160.3 | 0.128 | 41.9 | 327.4±7.9 | 11584.1.1 | 0.239±0.019 | 11505±648 | 4.0 |
| Belukha415 | 160.3-160.9 | 0.102 | 23.7 | 231.0±5.6 | 11585.1.1 | 0.144±0.041 | 15584±2365 | 1.9 |
| SLNS101 | 56.8-57.5 | 0.238 | 44.0 | 184.9±4.5 | 12458.1.1 | 0.972±0.016 | 227±131 | 1.9 |
| SLNS113 | 64.7-65.4 | 0.213 | 39.4 | 185.2±4.5 | 12459.1.1 | 0.942±0.016 | 484±137 | 1.7 |
| SLNS122 | 68.9-69.7 | 0.234 | 57.9 | 248.0±6.0 | 12460.1.1 | 0.773±0.010 | 2073±101 | 1.8 |
| SLNS127 | 71.8-72.5 | 0.252 | 57.8 | 229.7±5.5 | 12461.1.1 | 0.730±0.009 | 2527±101 | 2.2 |
| SLNS136 | 76.7-77.5 | 0.220 | 48.3 | 219.1±5.3 | 12462.1.1 | 0.657±0.009 | 3380±112 | 1.6 |
| SLNS139 | 78.9-79.6 | 0.208 | 48.1 | 230.8±5.6 | 12463.1.1 | 0.580±0.009 | 4381±131 | 1.8 |
| SLNS141-142 | 80.3-81.0 | 0.246 | 43.8 | 177.5±4.3 | 12464.1.1 | 0.550±0.010 | 4809±151 | 1.2 |
| CC237 | 126.0-126.7 | 0.208 | 28.5 | 136.6±3.3 | 12454.1.1 | 0.980±0.023 | 161±185 | 2.1 |
| CC244 | 130.2-130.8 | 0.167 | 21.7 | 129.8±3.1 | 12455.1.1 | 0.800±0.018 | 1789±185 | 1.4 |
| CC252 | 133.4-133.8 | 0.120 | 24.3 | 202.5±4.9 | 12456.1.1 | 0.546±0.016 | 4854±239 | 1.5 |

**Table 3** Calibrated WIO$^{14}$C and DO$^{14}$C ages using OxCal v4.3.2 with the Intcal13 radiocarbon calibration curve. Ages are given as the OxCal provided μ-age ± 1σ, which is the calibrated mean age accounting for the age probability distribution. In addition, calibrated ages derived when applying the OxCal sequence deposition model for further constraint are shown.

| Core section | WIOC Cal age (cal BP) | WIOC Cal age with sequence (cal BP) | DOC Cal age (cal BP) | DOC Cal age with sequence (cal BP) |
|---|---|---|---|---|
| CG110 | 1004±119 | 968±1049 | 464±235 | 403±196 |
| CG111 | 1224±103 | 1174±86 | 810±169 | 749±123 |
| CG112 | 1190±142 | 1292±103 | 901±176 | 947±139 |
| CG113 | 1889±138 | 1869±143 | 1222±153 | 1248±144 |
| Belukha412 | 8960±266 | 8954±268 | 10695±867 | 10686±865 |
| Belukha414 | 14796±782 | 14802±774 | 13646±893 | 13670±880 |
| Belukha415 | 22441±1107 | 22497±1107 | 20264±4073 | 20393±4033 |
| SLNS101 | 848±396 | 701±315 | 250±145 | 226±137 |
| SLNS113 | 1297±453 | 1255±331 | 480±131 | 505±111 |
| SLNS122 | 1769±514 | 1901±4301 | 2057±129 | 2056±129 |
| SLNS127 | 3175±679 | 3221±629 | 2585±125 | 2585±125 |
| SLNS136 | 6030±824 | 5426±620 | 3635±138 | 3636±137 |
| SLNS139 | 6026±820 | 6177±567 | 5014±191 | 5007±187 |
| SLNS141-142 | 6626±831 | 7081±689 | 5519±188 | 5531±176 |
| CC237 | 3051±703 | 2886±617 | 237±151 | 233±153 |
| CC244 | 4057±769 | 4210±713 | 1737±211 | 1738±212 |
| CC252 | 11000±1697 | 11017±1716 | 5580±294 | 5580±295 |

**Table 4** Estimate of the effect from in-situ $^{14}$C production on F$^{14}$C-DOC. For comparison, the measured F$^{14}$C offset between DOC and WIOC is also shown.

| Core section | Ice mass (g) | Carbon mass (µg) | Depth (m w.e.) | $P_o$ ($^{14}$C atom g$^{-1}$ ice yr$^{-1}$) | In-situ $^{14}$C (atoms) | In-situ F$^{14}$C-DOC offset | Observed F$^{14}$C DOC-WIOC offset | In-situ corrected F$^{14}$C-DOC | In-situ corrected DOC Cal age (cal BP) |
|---|---|---|---|---|---|---|---|---|---|
| CG110 | 171 | 18.9 | 55.8 | 328 | 1197 | 0.033±0.013 | 0.068±0.032 | 0.910±0.033 | 752±273 |
| CG111 | 207 | 25.5 | 56.3 | 328 | 1197 | 0.030±0.012 | 0.053±0.024 | 0.901±0.024 | 1045±207 |
| CG112 | 248 | 23.6 | 56.7 | 328 | 1197 | 0.038±0.015 | 0.037±0.026 | 0.889±0.026 | 1225±250 |
| CG113 | 246 | 29.5 | 57.0 | 328 | 1197 | 0.030±0.012 | 0.064±0.019 | 0.849±0.020 | 1546±208 |
| Belukha412 | 172 | 28.5 | 142.7 | 286 | 921 | 0.017±0.007 | -0.052±0.026 | 0.315±0.025 | 11271±902 |
| Belukha414 | 128 | 41.9 | 143.9 | 286 | 921 | 0.009±0.003 | 0.027±0.024 | 0.239±0.020 | 14096±964 |
| Belukha415 | 102 | 23.7 | 144.5 | 286 | 921 | 0.012±0.005 | 0.043±0.043 | 0.144±0.041 | 21571±4753 |
| SLNS101 | 238 | 44 | 47.9 | 345 | 2666 | 0.044±0.017 | 0.070±0.050 | 0.972±0.023 | 587±187 |
| SLNS113 | 213 | 39.4 | 54.4 | 345 | 2656 | 0.044±0.017 | 0.089±0.050 | 0.942±0.023 | 837±184 |
| SLNS122 | 234 | 57.9 | 58.1 | 345 | 2651 | 0.033±0.013 | -0.034±0.047 | 0.773±0.016 | 2483±210 |
| SLNS127 | 183 | 57.8 | 60.5 | 345 | 2647 | 0.026±0.010 | 0.029±0.047 | 0.730±0.014 | 2967±197 |
| SLNS136 | 220 | 48.3 | 64.7 | 345 | 2641 | 0.037±0.014 | 0.135±0.047 | 0.657±0.017 | 4264±304 |
| SLNS139 | 208 | 48.1 | 66.5 | 345 | 2638 | 0.035±0.014 | 0.058±0.046 | 0.580±0.016 | 5600±290 |
| SLNS141-142 | 246 | 43.8 | 67.7 | 345 | 2636 | 0.045±0.018 | 0.061±0.047 | 0.550±0.020 | 6323±363 |
| CC237 | 208 | 28.5 | 113.7 | 497 | 5371 | 0.120±0.046 | 0.275±0.054 | 0.980±0.052 | 1240±498 |
| CC244 | 167 | 21.7 | 117.6 | 497 | 5353 | 0.126±0.049 | 0.161±0.051 | 0.800±0.052 | 3509±799 |
| CC252 | 120 | 24.3 | 120.2 | 497 | 5341 | 0.080±0.031 | 0.231±0.051 | 0.546±0.035 | 7007±635 |

**Table 5** DO$^{14}$C dating results for near bedrock ice compared to results from previous studies (visualized in Figure 6).

| Site | Study | Core | Dating method | Depth above bedrock (m) | Age (cal BP) |
|---|---|---|---|---|---|
| Colle Gnifetti | this study | CG15 | DO$^{14}$C | (74.3 m below surface)[*] | 1248 ± 144 |
| | Jenk et al., 2009 | CG03 | WIO$^{14}$C | (73.5 m below surface)[#] | 1152 ± 235 |
| | Jenk et al., 2009 | CG03 | Model | (74.3 m below surface)[&] | $1160 \pm^{140}_{170}$ |
| | Jenk et al., 2009 | CG03 | WIO$^{14}$C | 0.6-0 | >15000 |
| | Jenk et al., 2009 | CG03 | Model | oldest ice estimate | $19100 \pm^{4800}_{4500}$ |
| Belukha | this study | B18 (saddle) | DO$^{14}$C | 0.5-0 | 20393 ± 4033 |
| | Aizen et al., 2016 | B03 (west plateau) | WIO$^{14}$C | 0.6-0.3 | 11015 ± 1221 |
| | Uglietti et al., 2016 | B03 (west plateau) | Model | 0.6-0 | 28500 ± 16200 |
| SLNS | this study | SLNS | DO$^{14}$C | 0.4-0 | 5531 ± 176 |
| | no previous results | | --- | --- | --- |
| Chongce | this study | Core 1 | DO$^{14}$C | 0.2-0 | 5580 ± 295 |
| | Hou et al., 2018 | Core 2 | WIO$^{14}$C | 1.2-0.8 | 6253 ± 277 |
| | Hou et al., 2018 | Core 2 | Model | oldest ice estimate | $9000 \pm^{7900}_{3600}$ |

[*]precise bedrock depth unknown at this coring site, [#]sampled depth being closest to depth sampled in this study (CG03 and CG15 drill sites only 16 m apart), [&]modeled age at same depth as sampled in this study.

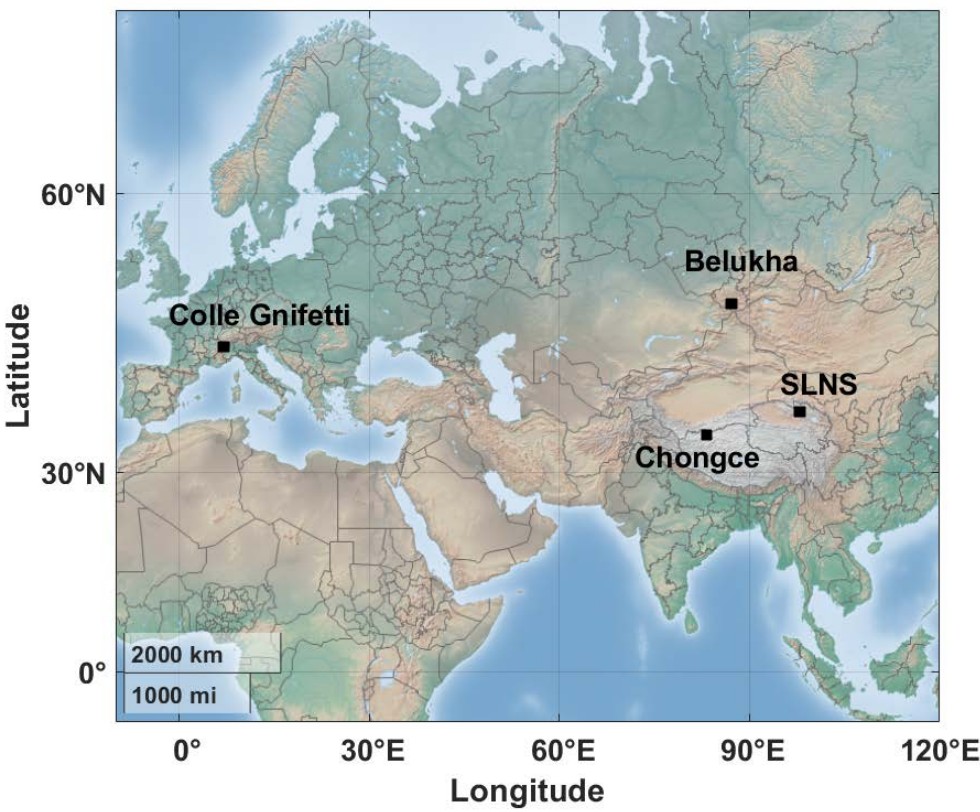

**Figure 1:** Location of the four glaciers Colle Gnifetti, Belukha, Chongce, and Shu Le Nan Shan (SLNS). Map made from Matlab R2019b geobasemap. Colle Gnifetti is located in the Monte Rosa massif in the Swiss Alps, Belukha glacier in the Altai mountain range, Russia, the Chongce ice cap on the northwestern Tibetan Plateau, China, and the SLNS at the south slope of the Shulenanshan Mountain, China.

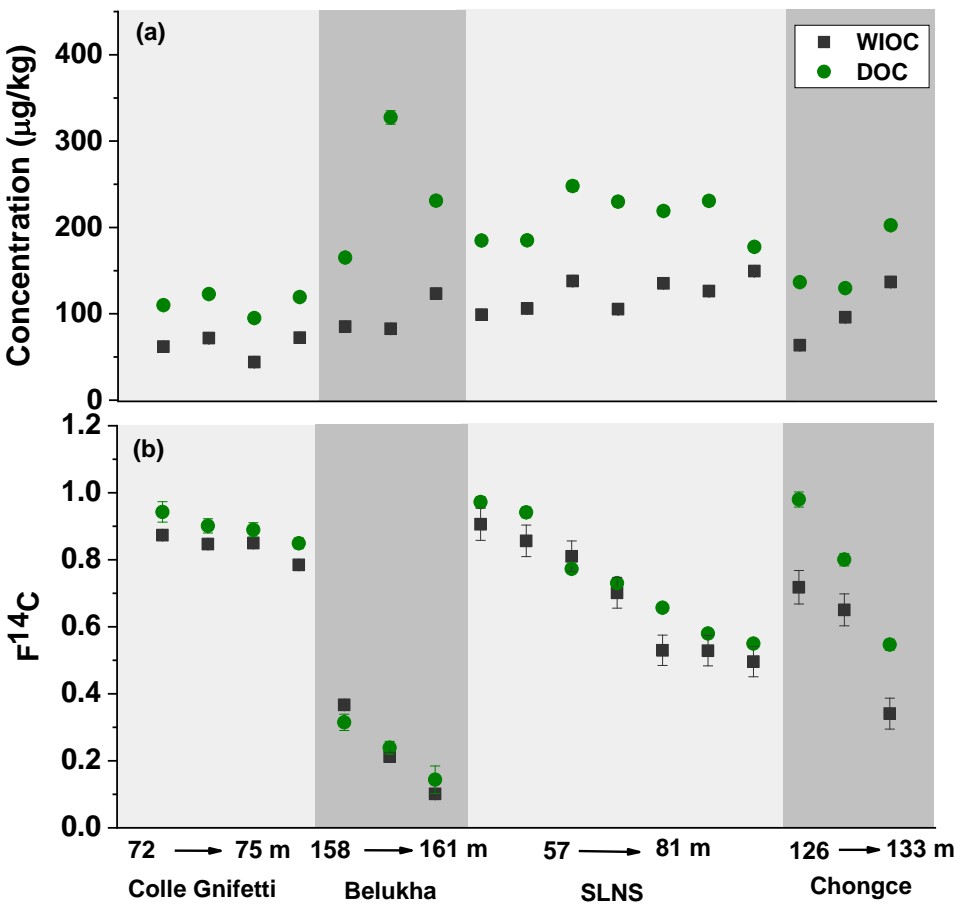

**Figure 2**: Comparison of results from the WIOC and DOC fractions for the studied four sites. (a) concentrations (b) F$^{14}$C. The error bars denote the overall analytical 1σ uncertainty.

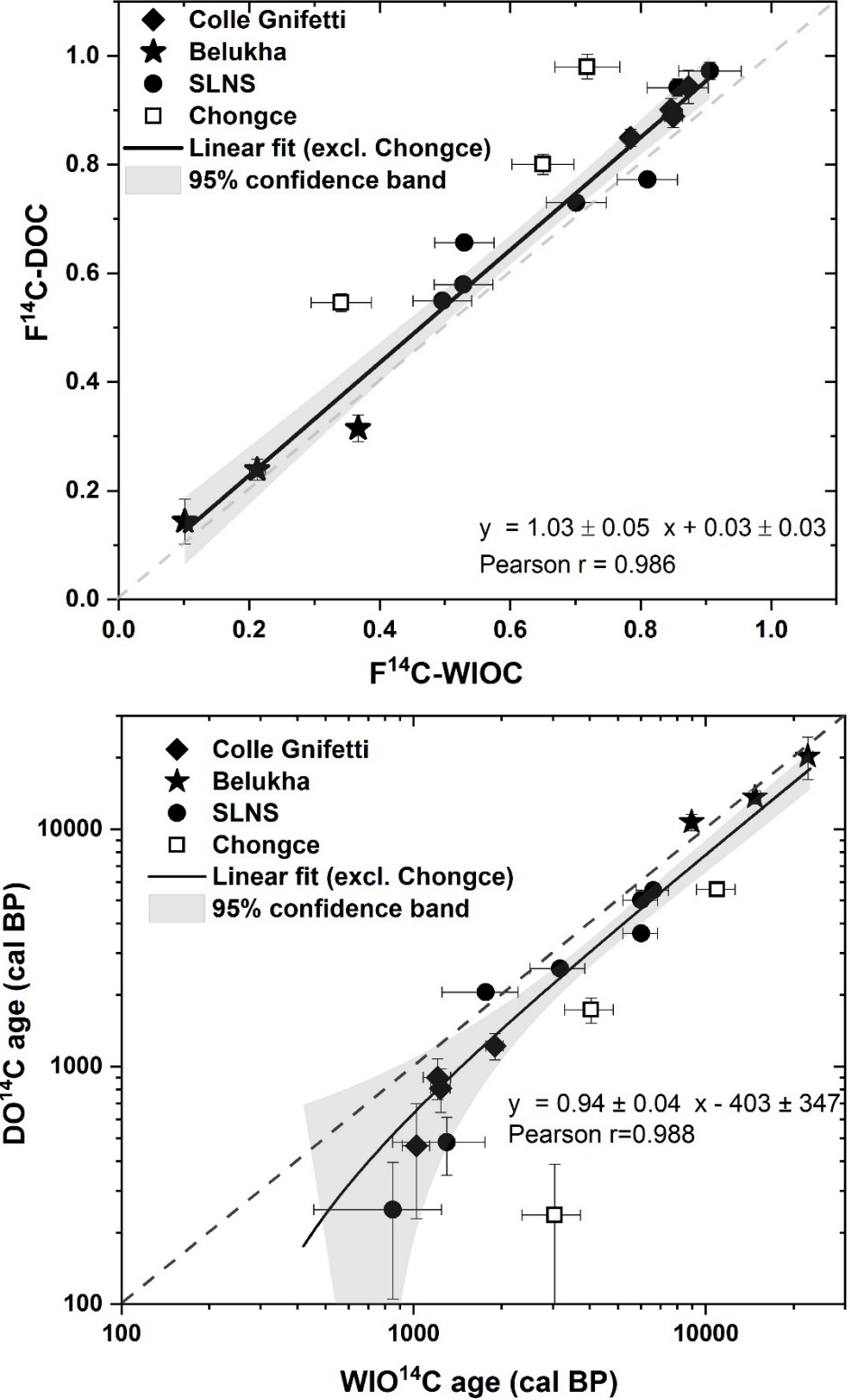

**Figure 3** Scatter plot showing the correlation between WIO$^{14}$C and DO$^{14}$C results for the four sites (see legend). In terms of F$^{14}$C (top) and calibrated ages (bottom). For the linear fit in both panels, the data from Chongce (open symbols) was excluded. Shaded areas indicate the 95% confidence band.

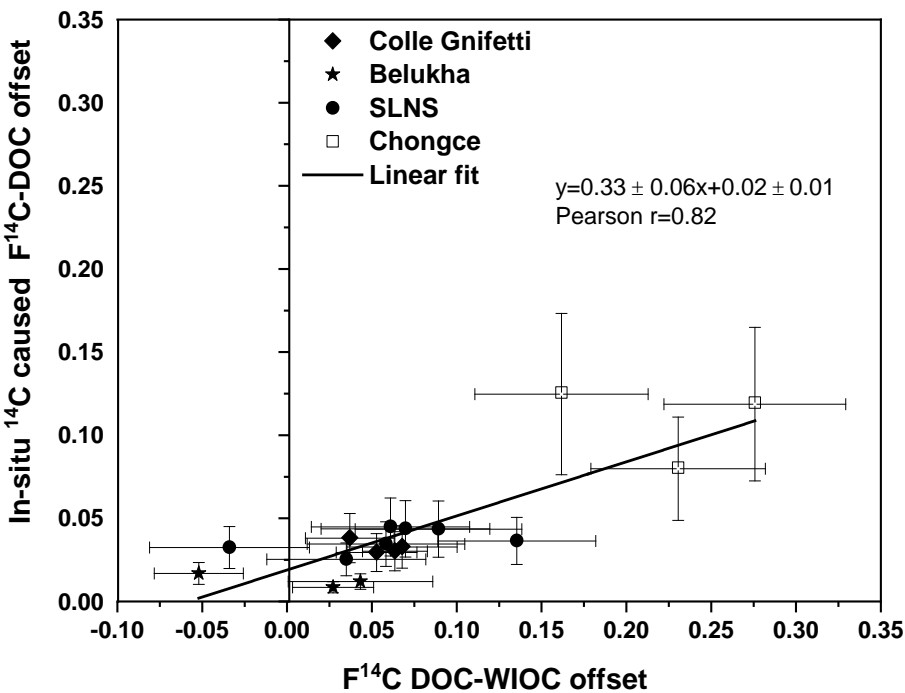

**Figure 4** Estimated in-situ $^{14}$C offset to F$^{14}$C-DOC plotted against the measured offset between F$^{14}$C of the DOC and WIOC fraction.

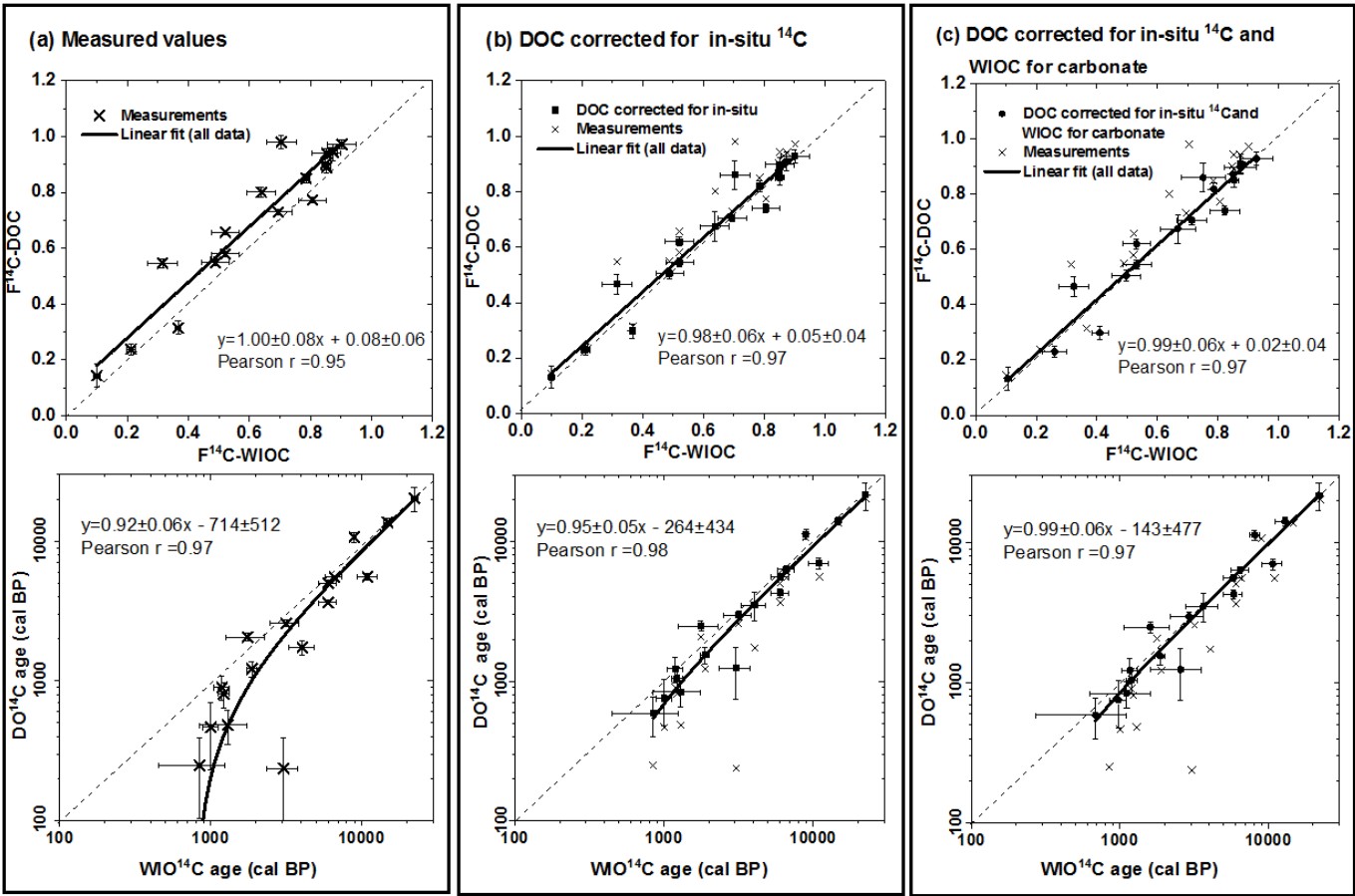

**Figure 5** Scatter plots showing the correlation between WIO$^{14}$C and DO$^{14}$C results for all samples. In terms of F$^{14}$C (top) and calibrated ages (bottom). (a) Measured values as shown in Figure 3 but with the linear fit applied to all data (Chongce included). (b) Same as panel (a), but DOC $^{14}$C results corrected for in-situ $^{14}$C contribution. (c) Same as panel (a), but DOC and WIOC $^{14}$C results corrected for in-situ $^{14}$C and accounting for potentially incompletely removed carbonate, respectively. An estimated average carbonate removal efficiency of 98±2 % was used here. Error bars in panel (a) and (b) reflect the propagated uncertainty of analysis and correction. In panel (b) and (c), measured values are shown as gray crosses.

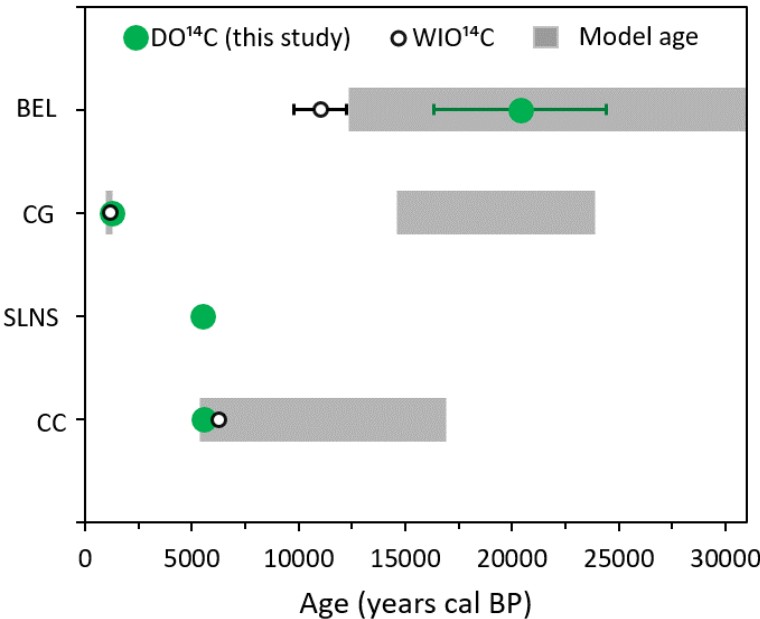

**Figure 6** Comparison of our DO$^{14}$C ages (not corrected for in-situ) with dating results from previous studies if available. For the four sites of Belukha (BEL), Colle Gnifetti (CG), Shu Le Nan Shan (SLNS) and Chongce (CC), DO$^{14}$C ages (green) and previously reported WIO$^{14}$C ages (open circles) for similar sampling depths are shown. Gray bars indicate previously modeled, $^{14}$C based bedrock age estimates (additionally for CG the modeled age for the bottom sampling depth of this study). Previously published data are from Uglietti et al. (2016) (BEL, West Plateau), Jenk et al. (2009) (CG), and Hou et al. (2018) (CC). See Table 5 for underlying data and details.