# Peer review of "Radiocarbon dating of alpine ice cores with the dissolved organic carbon (DOC) fraction"

_The Cryosphere, 2020_

## Referee Comment (RC1) · Anonymous Referee #1 · 10 Oct 2020

Review of
Radiocarbon dating of alpine ice cores with the dissolved organic carbon (DOC) fraction
by Ling Fang, Theo M. Jenk, Thomas Singer, Shugui Hou, Margit Schwikowski

The manuscript from Fang et al. investigates the possibility to use the dissolved organic carbon (DOC) fraction for 14C dating in high Alpine glacier ice. To do so the authors present an ice core sample set (17 ice core sections) taken from the deep parts of the high altitude Eurasian glaciers Colle Gnifetti, Belukha, Chongce, and Shule Nanshan, for which a direct 14C dating comparison between the water-insoluble organic carbon fraction (WIOC) and the DOC fraction was achieved for each sample.
It should be noted that "direct comparison" means that each of the 17 ice core sections samples was cut lengthwise and WIOC as well a DOC 14C was measured on each ice core section, i.e. on exact the same depth interval of the ice core.
Whereas the WIOC method is already well established, doubts were reported about suitability of the DOC fraction for 14C dating in an earlier study (May, 2009), what makes this study very challenging and important.
After a short description of the deployed WIOC and DOC sample preparation methods, WIOC and DOC concentration as well as the radiocarbon results are presented and discussed.
3 of the 4 sites show almost identically (not significantly different within the error) 14C ages for the corresponding samples, with a slight but systematic offset towards higher F14C values for DOC compared to WIOC. For one site (Chongce) this offset is enhanced. Since this latter site contains a high influence of dust in the ice the observed F14C DOC-WIOC offset is discussed by testing the hypothesis of an incomplete removal of carbonate during the WIOC sample preparation using the Ca2+ concentration in the samples as tracer for calcium carbonate

The paper is well structured and written in almost all parts and addresses an important scientific question, which is in the scope of TC. The study presents an up to this point unique data set which is suitable and convincing for the discussed topic and most of the conclusions made, and for which I would like to felicitate the authors.
The description of experiments and the presentation of the data as well as the discussion on the potential influence of incomplete removal of mineral dust on the WIOC sample preparation are except a few points (see my minor comments below) sufficiently complete and precise.
Therefore I think the manuscript should be published after a few minor and one major revision were made.

Apart from the minor points which are listed below, my major concern is that the paper lacks a more detailed discussion about the potential influence of in-situ produced 14C on the DOC radiocarbon content in high altitude glacier ice. Present state of the art in literature is that this effect makes the use of DOC unsuitable for 14C dating, at least at low accumulation, high altitude mountain site as the Colle Gnifetti (denoted CG in the following) (May 2009, Hoffmann 2016) from which samples are presented here.
At present state of the manuscript, the authors state:
"*The fact that none of the samples analyzed in this study (n=17) resulted in super modern F14C values (> 1) and the obtained significant correlation between the F14C of WIOC and*

*DOC (Sect. 3.2) and the resulting calibrated 14C ages (Pearson r = 0.988, p < .01, n=14, Figure S1) represent strong evidence against the previously suggested 14C in-situ production in the DOC fraction (May, 2009)."*

This argumentation could possibly be drawn referring to work from May (2009) only. Within this particular study 14C DOC measurements underlie relative high blank contributions and therefore a high uncertainty, and corresponding 14C POC data are likely influenced by altered soil and dust material incorporated in the ice due to high combustion temperatures. Thus the dataset of this study is very scattered and May (2009) could at that point only speculate on the existence of an in-situ 14C production on the DOC content in ice at CG.

However the work of Hoffmann (2016) proofed via neutron irradiation experiments that (i) the production of 14C in glacier ice and the incorporation into the DOC fraction is possible and (ii) gave a quantitative estimate of the DOC incorporated fraction of produced 14C in Alpine ice. Based on this, the study finally also details a way to calculate its influence on ice core samples from this site.

Since this work is not yet referenced in the present study, here a brief summary of what is outlined there:
In view that:
>1) The production of 14C atoms within the ice matrix by spallation of oxygen within the water molecule, induced by cosmic radiation (cited references: Lal et al., 1987; van de Wal et al., 1994; Mazarik and Reedy, 1995) is a known process.
>2) Potential 14C production in organic compounds as CO and CO2, but also in CH4 (cited: Kemp et al., 2002, Petrenko et al., 2009, 2013), as well as the possibility to hydrogenate the CO molecule to higher organic species (cited: Woon, 2002) are already reported in literature

Hoffmann (2016) performed the irradiation experiment mentioned above to confirm or not what is proposed in literature. The experiment showed that 14C in-situ production in DOC is a real process and suggests that between 11-25 % of the initially produced 14C atoms entered into the DOC fraction of Alpine Glacier ice.

On the base of that, as outlined by Hoffman (2016),
1) the theoretically produced number of 14C atoms for mid latitude glacier site at an altitude of 4500 m asl can be estimated as a function of accumulation rate and depth (based on literature data), and
2) the relative amount of 14C, which entered in the DOC fraction of Alpine Glacier ice can be estimated quantitatively from the neutron irradiation experiment.

Since this study exists, and the in-situ production in the DOC fraction would result in enhanced F14C fractions, I think it is really worth and necessary to take this effect into account. It should be discussed in this manuscript as partial or at least potential cause of the systematically observed DOC-WIOC difference, beside the hypothesis of the incomplete inorganic carbon removal within the WIOC sample preparation (which

surely is also a good candidate for the observed offset in case mineral dust is present in the samples).

Having been curious myself on the order of magnitude this effect would have on the DOC 14C values measured in this study, I did a back-of-the-envelope calculation of the in-situ effect by applying the calculations of Hoffmann (2016) on the CG samples of this study.

Accumulation rate and depth in water equivalent of CG15 are not given in the manuscript. Since however similar ages were found at similar depths in the cores CG15 and CG03 (see table 3 and section 4.3) the respective data from the CG03 (drilled in 2003 almost directly at the saddle point of CG, Jenk et al., 2009) were used for the estimation. As the ice in the deeper part of the C15 core probably originates from upstream the drill site, i.e. from a position on the north flank of the CG, where the accumulation is lower (e.g. Licciulli et al., 2020), and since the accumulation rate is one of the driving factors of the magnitude of the in-situ production in the estimation, calculations for different accumulation rates were carried out. The mean of uppermost 30 years of CG3 (0.47 mwe/yr) was used, and additionally two values (0.25, and 0.12 mwe/yr), which are in the order of magnitude of what is found upstream in the north flank of the CG. Since all four samples were taken from about the same depth, and had the same sample and carbon masses, mean values of (220g, 24ugC, and 56.75 mweq) were used in the calculation. As relative fraction of 14C, which entered in the DOC fraction 15% were assumed.

The estimation resulted in potential F14C offsets of 0.025, 0.047, and 0.096 for the assumed accumulation rates of 0.47 mwe/yr, 0.25, and 0.12 mwe/yr, respectively, which fits quite well with the observed offset within this study (0.055±0.014).

Therefore, as stated above, a discussion of the in-situ production of 14C influencing the DOC 14C dating should not be neglected but done here. It would also significantly improve the scientific output of the manuscript. In addition, in view of the expected results, all existing studies on this topic would become conclusive and an important gap of knowledge in literature could be closed.

Minor comments:

Line 34-36: it would be good if you could give an idea of how much ice would be needed (inclusive the lost during decontamination) for an Antarctic sample (see also my comment on line 389).
In addition, be aware and mention that the potential in-situ effects will be much stronger in Polar Regions than in the high altitude sites in mid-latitudes, since the neutron flux and thus production rate is higher and accumulation rates are generally lower there.

Line 48-50: … Ice flow models, which are widely used to retrieve full depth age scales (e.g. Nye, 1963; Bolzan, 1985; Thompson et al., 2006), also fail in the deepest part of high-alpine glaciers due to the complex bedrock geometry. ….

Please clarify or revise this sentence. To my knowledge the high model uncertainty in the deepest part of the glacier (which includes for me the deepest 5-10m above bedrock) its not only due to the bedrock geometry, but rather to the uncertainties in the assumptions needed to be made to constrain the model and which include beside the bedrock geometry also mass balance upstream, equation of temperature depended shear stress, steady state conditions.

Line 59-62: Samples of >10 µg WIOC can be dated with reasonable uncertainty (10-20%), requiring less than 1 kg of ice from typical mid-latitude and low-latitude glaciers (Jenk et al., 2007; Jenk et al., 2009; Sigl et al., 2009; Uglietti et al., 2016).

Please include also the study of Hoffmann et al., 2018, in which 14C dating on the WIOC franction was achived with an other sample preparation setup. Be also please more precise on the sample and carbon mass needed to achieve such an uncertainties of 10-20%. It seems that Hofmann et al. 2018 achieves this uncertainty with an ice mass <500g and a carbon mass of <10 µgC. Also it would be good to mention whether the AMS or the sample preparation error dominates the uncertainty.

Line 79-81: In view of the analytical precision achievable with this method, the turn-over time from atmospheric CO2 to deposited aerosol is negligible (Fang et al., in prep.).

I am not sure if I got the meaning here.
Do you mean the analytical uncertainty, which results in an age error, which is much higher than the turn-over time?

Line 93- 95: … possible mechanisms of 14C in-situ formation in organic compounds seem far less likely and have not been investigated to date…

This sentence needs to be revised (see my major comment), since 14C in-situ formation in DOC of high Alpine glacier ice was investigated.

Line 103-104: ….allowing 14C analysis on samples with DOC concentrations as low as 25 µg/kg ….

I guess this assumption is made in view of the required carbon mass needed for 14C sample preparation and/or measurements. If true please mention that and change the sentence to something like:
The system can handle samples with volumes of up to ~350 mL. To achieve a minimal carbon mass required for 14C sample ……?  A minimal DOC concentration of 25 µg/kg is needed.

Line 116 – 133: It might be worth to summarize the meta data on the ice cores and samples listed here in a table (including geographic coordinates of the drill site, ice core lengths, accumulation rate at the drill site, sampled depths in this study, ... the mountain range and reference to study in which more meta data on the cores are given).
In any case at least the accumulation rate and the references to further meta data of the different cores should be added in the text.

Line 185-187: ... and procedure blanks (1.26±0.59 µgC with F14C of 0.69±0.15 for WIOC samples and 1.9±1.6 µgC with a F14C value of 0.68±0.13 for DOC samples)...

The way the WIOC and DOC procedure blanks were made and the frequency or number of blanks achieved during the analysis of this study should be given.

Line 254 – 259: The fact that.... to ... (May, 2009).
In view of my request to discuss the potential bias due to 14C in-situ production by calculating its effect, these lines should be deleted.

Line 266 – 269: For DOC concentrations observed in this study, an initial ice mass of about 250 g was required, with about 20-30 % of the ice being removed during the decontamination processes inside the DOC set-up, yielding ~200 g of ice available 269 for final analysis.

This sentence should be moved to Section 2 in the paragraph, which starts in line 156.

Line 271:
Please specify here that the reduction of the sample mass in DOC refers to the WIOC method used at the PSI.

Line 276: Please add a section (4.2 or 4.3) on the "Potential contribution of 14C in-situ production to 14C of DOC" (see major comment)

Line 281-182: please change to something similar to:
... upper parts of the Chongce Cores 2 and 4, less than 2 and ~6 km away from Core 1, (measured with the same analytic device as used here), ...
...

Line 326-329: ... For final calibration of 14C ages, most of those earlier studies took advantage of the assumption of sequential deposition in the archive, which seems very reasonable considering the deposition of annual snow layers on top of each other on the glacier surface.....

Please be more prudent here and revise this sentence since several studies emphasized that a sequential deposition in the archive of high Alpine glaciers is not evident (a least in the case for CG, see Jenk et al., 2009, Hoffmann et al., 2018, Bohleber, 2019).
E.g. Bohleber 2019 wrote:
*"… as already noted by Jenk et al. (2009), the finding of a continuous age-depth relation in the deep core parts is not a priori to be expected (e.g., as strong shear could potentially decouple the deformation of the basal ice frozen to bed from its adjacent top layer, which would be reflected in a hiatus in the age-depth relation). In fact, the 14C profile obtained by Hoffmann et al. (2018) for a core located on CG's north-facing slope (with significant bedrock inclination, cf. the saddle location of the core investigated by Jenk et al., 2009) revealed a localized discontinuity in 14C ages…"*

Therefore I propose to argument like that:
1) Despite the fact that a sequential deposition in the archive is not evident in the deepest layers … (references…)
2) but in view that in case of relatively large analytical uncertainties compared to the age difference of the samples, the sequential deposition model can moderately constrain the probability distribution of the calibrated age ….

=> The sequence model was used but results were compared using the conventional calibration approach. …

Line 323: 4.3 DO14C ages in the context of published chronologies
In view of what is discussed in this paragraph I recommend to change the title to:
DO14C ages in the context of published near bedrock ice ages

Line 351 – 356, Table 5 and Figure 4:
1) to be complete for the CG site, please add also near bedrock ice age data obtained by Hoffmann et al., 2018 on an CG ice core (KCC) located on the north facing slope of the glacier, to the compilation of near bedrock ice ages. In the latter study the age difference of near bedrock ice between CG03 and the KCC is discussed, and might worth to be mentioned that here.
2) As already mentioned, the comparison of absolute depths between CG03 and CG15 leads to assume that both ice cores were drilled at the same location of CG. If true add this information in line 116

Line 389 … This new dating method opens up new fields for radiocarbon dating of ice for example from remote or Polar Regions, where concentrations of organic impurities in the ice are particularly low ….
To illustrate this statement, please give an estimation of how much ice (in g or kg inclusive the ice mass which is needed for decontamination) would be necessary to achieve a 14C dating on an ice sample. Typical DOC concentrations from Antarctic ice with an for 14C dating accessible age ( < 10 ppb) are given e.g. in Legrand et al., 2013. In addition as already stated in my comment to line 34-36, you should mention the potential influence of the 14C in-situ production which is expected to be enhanced

compared to high altitude sites in mid latitudes, and will thus result in an enhanced age uncertainty.

Literature cited in review:

Bohleber, Pascal. (2019). Alpine Ice Cores as Climate and Environmental Archives. 10.1093/acrefore/9780190228620.013.743.

Hoffmann HM. 2016. Micro radiocarbon dating of the particulate organic carbon fraction in Alpine glacier ice: method refinement, critical evaluation and dating applications [PhD dissertation]. Combined Faculties for the Natural Sciences and for Mathematics of the Ruperto-Carola University of Heidelberg. (available at: http://archiv.ub.uni-heidelberg.de/volltextserver/20712/ )

Hoffmann, H., Preunkert, S., Legrand, M., Leinfelder, D., Bohleber, P., Friedrich, R., & Wagenbach, D. (2018). A New Sample Preparation System for Micro-14C Dating of Glacier Ice with a First Application to a High Alpine Ice Core from CG (Switzerland). Radiocarbon, 60(2), 517-533. doi:10.1017/RDC.2017.99

Jenk, T. M., Szidat, S., Bolius, D., Sigl, M., Gaeggeler, H. W., Wacker, L., Ruff, M., Barbante, C., Boutron, C. F. and Schwikowski, M.: A novel radiocarbon dating technique applied to an ice core from the Alps indicating late Pleistocene ages, J Geophys. Res. Atmos., 114, 457 https://doi.org/10.1029/2009JD011860, 2009.

Legrand, M. ; Preunkert, S. ; Jourdain, B. ; Guilhermet, J. ; Fain, X. ; Alekhina, I. A. ; Petit, J. R.: Water-soluble organic carbon in snow and ice deposited at Alpine, Greenland, and Antarctic sites: a critical review of available data and their atmospheric relevance. In: Climate of the Past 9 (2013), pp. 2195–2211

Licciulli, C., Bohleber, P., Lier, J., Gagliardini, O., Hoelzle, M. and Eisen, O.: A full Stokes ice-flow model to assist the interpretation of millennial-scale ice cores at the high-Alpine drilling site Colle Gnifetti, Swiss/Italian Alps, J. Glaciol., 66, 35-48, https://doi.org/10.1017/jog.2019.82, 2020.

May B. 2009. Radiocarbon microanalysis on ice impurities for dating of Alpine glaciers [PhD dissertation]. Institut für Umweltphysik, Heidelberg University.

Kemp, W.J.M. & Alderliesten, C. & van der Borg, Klaas & Jong, A.F.M. & Lamers, Robert-Jan & Oerlemans, J. & Thomassen, M. & Wal, R.S.W.. (2002). In situ produced 14C by cosmic ray muons in ablating Antarctic ice. Tellus, Series B: Chemical and Physical Meteorology. 54. 10.1034/j.1600-0889.2002.00274.x.

Lal, D. ; Jull, A.J.T ; Donahue, D. J. ; Burtner, D. ; Nishizumi, K.: In situ cosmogenic 3H, 14C and 10Be for determinig th net accumulation and ablation rates of ice sheets. In: Journal of Geophysical Research, Solid earth 92 (1987), Nr. B6, pp. 4947–4952.

Mazarik, J. ; Reedy, R. C.: Terrestrial cosmogenic-nuclide pro- duction systematics calculated from numerical simulations. In: Earth and Planetary Science Letters 136 (1995), pp. 381–395

Petrenko, V. V. ; Severinghaus, J. P. ; Smith, A. M. ; Riedel, K. ; Baggenstos, D. ; Harth, C. ; Orsi, A. ; Hua, Q. ; Franz, P. ; Takeshita, Y. ; Brailsford, G. ; Weiss, R. F. ; Buizert, C. ; Dickson, A. ; Schaefer, H.: High-precision 14C measurements demonstrate production of insitu cosmogenic 14CH4 and rapid loss of insitu cosmogenic 14CO in shallow Greenland firn. In: Earth and Planetary Science Letters 365 (2013), pp. 190–197

Petrenko, V. V. ; Smith, A. M. ; Brook, E. J. ; Lowe, D. ; Riedel, K. ; Brailsford, G. ; Hua, Q. ; Schaefer, H. ; Reeh, N. ; Weiss, R. F. ; Etheridge, D. ; Severinghaus, J. P.: 14CH4 Measurements in Greenland Ice: Investigating Last Glacial Termination CH4 Sources. In: Science 324 (2009)

Wal, R. S. W. van de ; Roijen, J. J. van ; Raynaud, D. ; Borg, K. van der ; Jong, A. F. M. de ; Oerlemans, J. ; Lipenkov, V. Y. ; Huybrechts, P.: From 14C/12C measurements towards radiocarbon dating of ice. In: Tellus 46B (1994), pp. 94–102

Woon, D. E.: Modeling gas-grain chemistry with quantum chemical cluster calculations. I. heterogeneous hydrogenation of CO and H2CO on icy grain mantles. In: The Astrophysical Journal 569 (2002), pp. 541–548

---

## Referee Comment (RC2) · Anonymous Referee #2 · 10 Oct 2020

Review of Fang et al

This manuscript presents the first results of a technique that utilizes 14C of DOC to date alpine ice cores. The basal sections of alpine ice cores are difficult to date because of high degree of ice thinning (making layer counting impossible) and complex ice flow. The approach used in this study is very analytically challenging, and in my opinion the method has been carefully developed and tested. The analytical precision on the F14C values is impressive considering the small sample sizes. Considering that this approach requires smaller ice samples ($\approx$250 g) than the earlier approach developed by the same group that uses insoluble organic carbon, this now seems like the most promising technique for dating alpine basal ice. Overall, I think that this is an exciting study that is in principle well suited for The Cryosphere. However, I think the study and

manuscript also have some weaknesses that should be addressed.

Major Comments:

One of the major goals of this study / manuscript is validation of the 14C-DOC technique. In my opinion the manuscript doesn't fully achieve this. The main approach for this evaluation is comparison with the WIOC-14C results. But those results seem to be affected (to varying degrees) by 14C interference from carbonate dust in the samples. Are the authors able to measure a few samples from a layer-counted Greenland ice core, for example, to provide a more robust validation? I realize that this may be difficult, both because of lower DOC concentrations and ice availability, but perhaps a core from a coastal ice cap such as Renland could be a good target?

I think it would be valuable to provide a more complete analysis of the overall dating uncertainties. If 14C-WIOC is the benchmark measurement that is being used for validation of 14C-DOC, then the uncertainties in the 14C-DOC ages need to fully reflect the uncertainties associated with us 14C-WIOC (see more on this below). Alternatively, if the authors consider 14C-DOC to be an inherently superior approach (as compared to 14C-WIOC), then a more clear argument needs to be made for this.

I think the uncertainties associated with the correction for carbonate dust (for WIOC-14C) need to be more thoroughly considered. The authors provide some helpful discussion of this in the supplement (starting on line 33), but I'm not convinced that the uncertainties are fully accounted for. For example, it seems to me that F14Ccarb could in principle range from 0 to 1 depending on the source of the carbonate. One could imagine a situation with seasonally-drying lakes in arid regions, for example, where the carbonate dust at the surface would be close to modern in its 14C signature. The C / Ca ratio in dust derived from dolomite would be twice as large as what is being used in Supplement equation 2. The effect of these additional uncertainties may be visible in figure 3b – while the correction makes the Chongce samples look more reasonable, two of the Belukha samples now fall off the trend.

Minor Comments: Line 73. For water-soluble organics sourced from biomass burning, there may be an age offset due to the older ages of the burned material. While this is probably small compared to the measurement uncertainties, it would still be worth mentioning briefly. Similar comment for organics sourced from oceanic emissions (affected by ocean radiocarbon reservoir effect).

Line 191. Why is the Libby half-life of 14C being used here instead of the more accurate value of 5730 yrs?

Figure 3b legend. Use a label that's more descriptive than "corr" for the corrected data; perhaps just say "corrected results"

Line 278. "As described in Section 3.2, no significant difference between F14C of DOC and WIOC was observed for the ice samples from Colle Gnifetti, Belukha and SLNS (Figure 3)." This is incorrect. Based on the figure, the differences seem significant at the 1-sigma level for several samples, and at the 2-sigma level for at least 1 sample.

Table 5 / Figure 4 and associated text. I think that the discussion of the limitations of this comparison to the previous age estimates should be expanded to provide some more detail / caveats associated with the comparison. For example, how close were the CG and Chongce cores to each other / do we even expect the basal ice to be of similar age? Is the comparison at Belukha still meaningful given the different core locations?

While I think that the authors' conclusion that there is no evidence that in situ 14C is affecting the 14C-DOC measurements is likely correct, the authors should do a better job of supporting this conclusion. For example, how can you be certain that the higher 14C-DOC as compared to 14C-WIOC in most samples is not due to in situ 14C? The largest offsets are observed at Chongce, which has the highest altitude and therefore should in principle have the highest in situ 14C production rates.

---

## Author Comment (AC2) · 23 Nov 2020

**Author's response to referee comments on: "Radiocarbon dating of alpine ice cores with the dissolved organic carbon (DOC) fraction" by Fang et al.,**

*Correspondence to:* Theo M. Jenk (theo.jenk@psi.ch)

We would like to thank the reviewers for their constructive comments that helped us to improve the accuracy of our evaluation of the potential of DOC for radiocarbon dating. Our responses to their comments are in blue.

**Anonymous Referee #2**

This manuscript presents the first results of a technique that utilizes 14C of DOC to date alpine ice cores. The basal sections of alpine ice cores are difficult to date because of high degree of ice thinning (making layer counting impossible) and complex ice flow. The approach used in this study is very analytically challenging, and in my opinion the method has been carefully developed and tested. The analytical precision on the F14C values is impressive considering the small sample sizes. Considering that this approach requires smaller ice samples (_250 g) than the earlier approach developed by the same group that uses insoluble organic carbon, this now seems like the most promising technique for dating alpine basal ice. Overall, I think that this is an exciting study that is in principle well suited for The Cryosphere. However, I think the study and manuscript also have some weaknesses that should be addressed.

Major Comments:
One of the major goals of this study / manuscript is validation of the 14C-DOC technique. In my opinion the manuscript doesn't fully achieve this. The main approach for this evaluation is comparison with the WIOC-14C results. But those results seem to be affected (to varying degrees) by 14C interference from carbonate dust in the samples. Are the authors able to measure a few samples from a layer-counted Greenland ice core, for example, to provide a more robust validation? I realize that this may be difficult, both because of lower DOC concentrations and ice availability, but perhaps a core from a coastal ice cap such as Renland could be a good target?

As we discuss in the manuscript, the carbonate effect on WIOC is small and within the range of the analytical uncertainty. The ages of ice obtained with WIOC were validated before by comparison with independently dated ice (by annual layer counting or conventional [14]C dating of microfossils contained in the ice) and this was published and we cite that (Uglietti et al., 2016). We therefore don't see the necessity to measure further samples from Greenland to validate the WIOC dating method used as benchmarker in this study.

I think it would be valuable to provide a more complete analysis of the overall dating uncertainties. If 14C-WIOC is the benchmark measurement that is being used for validation of 14C-DOC, then the uncertainties in the 14C-DOC ages need to fully reflect the uncertainties associated with us 14C-WIOC (see more on this below). Alternatively, if the authors consider 14C-DOC to be an inherently superior approach (as compared to 14C-WIOC), then a more clear argument needs to be made for this. I think the uncertainties associated with the correction for carbonate dust (for WIOC-14C) need to be more thoroughly considered. The authors provide some helpful discussion of this in the supplement (starting on line 33), but I'm not convinced that the uncertainties are fully accounted for. For example, it seems to me that F14Ccarb could in principle range from 0 to 1 depending on the source of the carbonate. One could imagine a situation with seasonally-drying lakes in arid regions, for example, where the carbonate dust at the surface would be close to modern in its 14C signature. The C /Ca ratio in dust derived from dolomite would be twice as large as what is being used in Supplement equation 2. The effect of these additional uncertainties may be visible in figure 3b – while the

correction makes the Chongce samples look more reasonable, two of the Belukha samples now fall off the trend.

Regarding the main issues raised by both reviewers, we do understand the concerns and would like to thank for the careful evaluation of the manuscript. While details can certainly be discussed, we however are a bit surprised that the reviewer here asks for an even more detailed analysis of the overall dating uncertainties. This considering the fact, that we discuss discrepancies, which are barely statistically significant (below the analytical detection limit for 4 out of 3 sites). Since the method of $^{14}$C-WIOC dating has been validated previously (see Uglietti et al., 2016) we also think that it is justified to use this as a benchmark. Anyway, in the revised version of the manuscript, we will include the valuable suggestions of the two reviewers to even further improve this in-depth discussion (see comments related to in-built ages and in-situ production). Consequently, with the new consideration of potential in-situ contribution to DO$^{14}$C, this fraction will no longer be considered to be the superior approach. Instead, for both fractions, we are confident to be able to provide more precise and accurate guidelines about potential limitations in the accuracy for both approaches.

In the new calculation about in-situ production, we find about 50% of the offset between F$^{14}$C DOC-WIOC can be explained by in-situ production, see related comment. Although numbers of carbonates contribution to WIOC will thus change, the related modeling approach will still be part of the manuscript. First, we would like to stress, that this modeling results should not be viewed as a mean for correction of WIOC F$^{14}$C results. Instead, the aim was to test the hypothesis that a less than 100% efficient carbonate removal procedure could potentially explain the observed offset between F$^{14}$C of WIOC and DOC with the required level of efficiency being plausible (high and only slightly less than 100%). As we stated, the future aim would be to improve the carbonate removal process, not to correct WIO$^{14}$C for a potential carbonate bias. Under this aspect, we agree with the reviewer, that the selection of the parameter space for F$^{14}$C$_{carb}$ and the carbonate-to-calcium ratio is critical. This is what we already stated in the supplement. Our opinion is that therein (lines 33-42 as pointed out by the reviewer), the range of possible F$^{14}$C values for carbonates or carbonate-to-calcium ratios was already reasonably explored. We discussed the robustness of the modeling results, provided an idea of the associated uncertainty by exploring the parameter space. However, not intending to apply a correction for (a potential) carbonate bias, we thereby did not focus to precisely quantify uncertainties but considered an evaluation of those to be sufficient for determining if the model results are robust in terms of the estimated removal efficiency required for explaining the observed offset. For explanation in this response only, we will summarize below the reasoning behind our approach and point out what was already addressed in the supplement:

(1) The carbonate removal efficiency by the acidification step very likely depends on the source and transport of the mineral dust (geological form, particle size) affecting the solubility. Thus, tuning the model for a common carbonate removal efficiency ($x_{eff}$) instead of allowing this parameter to vary for individual sites will likely will not yield the best possible approximation between the observations and the modeled data as reflected in Figure 3b. In the Supplement, line 38-45, we thus discussed results with a model set-up allowing the carbonate removal efficiency to be different for the individual sites (within +- 4% for all sites).This showes, that the estimated value for $x_{eff}$ may not be a precise, best value for each site but a robust average estimation.

(2) As pointed out by the reviewer, the abundance of carbonates in their different geological forms is likely different for each site (similar to point (1) dependent on the source region) and thus the value for the C/Ca ratio is likely not a fixed single value. However, we do not know what value would be most appropriate for each of the sites. Allowing this parameter

to be free in the model, i.e. allowing it to tune to a "best" value for each individual site would thus be speculative and over-tuning of the model. Anyhow, please note, that in the Supplement we did some evaluation of changes to this parameter by using a different value for this ratio (0.8 instead of 0.5), which again provided an estimate about the robustness/uncertainty of the final result for $x_{eff}$.

(3) $F^{14}C_{carb}$ very likely differs dependent on the site (again relates to the source). Best, $F^{14}C_{carb}$ would be selected individually for each sample, based on the difference in age of the contemporary atmosphere to the assumed age of the source carbonate at that time. But also here, the available literature is sparse and without speculation we cannot assume "precise" values. The value suggested by the reviewer ($F^{14}Ccarb = 1$) would certainly not be a reasonable value to assume, it reflects the year 1950 AD. We think his idea was that at the time when a certain snow/ice layer now at some depth in the glacier was on the surface, mineral dust (carbonates) with an $F^{14}C$ being contemporary at that time were deposited as impurities in this layer. In this case the upper limit for $F^{14}Ccarb$ would be equal to the $F^{14}C$ corresponding to the age of this layer. It is clear, that a close to contemporary $F^{14}C$ would not introduce a bias and carbonates then could not explain the observed offset between DOC and WIOC. However, layers from "input from seasonally-drying lakes in arid regions" may occur as individual, special events but as such do not represent the norm (also for some sites, this can be excluded as a possibility entirely). Anyway, in the Supplement we already calculated using a different value for $F^{14}C_{carb}$ (0.05).

To conclude, we are very much aware that if the intention would be to model the observed offset as closely as possible to come up with a correction, the model-set up and parameter space could be explored in even more detail (note, that in principle, by tuning each parameter for individual sample, we can reproduce the offset to nearly 100%). However, in our opinion this would only be justified if knowing with absolute certainty that a contribution from carbonate carbon to $F^{14}C$ of WIOC exists and one could start to quantify and investigate the individual parameter values in detail. Again, we would like to point out that we look at discrepancies very close to the detection limit (or even below). Here we can thus only provide evidence that such an effect exists, causing a systematic offset in the direction observed. As mentioned in the beginning, now being aware that in-situ $^{14}C$ contribution to DOC cannot be completely ruled out, the potential "carbonate bias" becomes even smaller (i.e. the removal efficiency even higher) than what we estimated before.

Minor Comments:
 Line 73. For water-soluble organics sourced from biomass burning, there may be an age offset due to the older ages of the burned material. While this is probably small compared to the measurement uncertainties, it would still be worth mentioning briefly. Similar comment for organics sourced from oceanic emissions (affected by ocean radiocarbon reservoir effect).
We agree with the reviewer that there is an in-built radiocarbon age from old carbon reservoirs to be assumed, actually both, for the DOC and WIOC fractions. Depending on the region, the mean reservoir age of these potential sources is likely variable. For the reservoir age from biomass burning the mix of tree ages in the forests along the moisture source/transport pathways may differ and for organics sourced from oceanic emissions, both continentally of the site and the strength of upwelling/mixing with ocean deep waters in the source region will have an effect. In any case, the reviewer is correct when assuming that these potentially in-built ages are small, compared to the analytical uncertainty. In the study of Uglietti et al. (2016) where WIO$^{14}$C ages were compared to ages derived by independent methods, no bias was identified.

The mixed age of trees in Swiss forests today is estimated to be slightly less than 40 years (Mohn et al., 2008). Back in time, prior to extensive forest management (e.g. cutting of trees reaching a certain age, removing of dead tree logs from the ground), the mixed age of trees in Europe was likely older. However, the most abundant European tree species reach ages of less than 100 years only. Zhang et al. (2017) found a relatively young stand age for Chinese forests mostly due to the large proportion of newly planted forests (0–40 years old), which are more prevailing in south China. Older forests (stand age>60 years old) are more frequently found in east Qinghai-Tibetan Plateau and the central mountain areas of west and northeast China, where human activities are less intensive. The oldest mean stand age was found to be 136 years old, and the youngest 18 years. For old-growth forests, e.g. in British Columbia (USA) with very limited human forestry management (old, dead tree logs still present), average in-built ages in charcoal of around 200-300 years were determined (Gavin, 2001). Note that hard-wood trees in this region can become exceptionally old (several hundreds of years) but maybe can assumed to be also representative for Siberian forests.

While charcoal is associated 100% with biomass burning (as can be assumed also for EC), WIOC is estimated to equally originate from direct biogenic emissions (similar in $^{14}$C content as the contemporary atmosphere, i.e. atmospheric $CO_2$) and biomass burning with around 50% (before the use of fossil fuels, Minguillon et al., 2011). For biogenic DOC, May et al. (2013) estimated a turnover-time of around 3-5 years for a study site in Switzerland. This corresponds to a contribution of around 20% from biomass burning which can easily be derived by using the mixed age of trees in Swiss forests of 40 years (see above). Note that a living tree is build up by rings with the outermost, youngest ring having contemporary age and thus, if a living tree of e.g. 40 years is completely burned, the released carbon will have an average $F^{14}$C corresponding to a mean age of 20 years only. Biomass burning also includes contribution from burning of grasslands and/or bushfires (young in age). Based on the above, a potential in-built age from biomass burning to the WIOC and DOC fraction can be estimated. When considering a value of 150±100 years for the mean age of burned material (aged wood plus grass and bushes), the potential in-built age from biomass burning for WIOC and DOC results with around 75±50 years and 30±20 years, respectively.

Collle Gnifetti is located more than 850 km away from the Atlantic Ocean and 250 km from the Mediterranean Sea (not in the main transport trajectory) and the other sites are even much more continental. For ice cores from the Alps the concentrations of methanesulfonate (MSA), a marine organic tracer, are more than one order of magnitude lower compared to the terrestrial organic tracers (e.g. formate, acetate).

In conclusion, regarding the analytical uncertainty of around 10-20% for radiocarbon dating by WIOC and DOC, these in-built ages are insignificant, at least for samples being older than a few hundred years. We will add a sentence or two to the manuscript as suggested and briefly summarize what we outlined in detail here.

Line 191. Why is the Libby half-life of 14C being used here instead of the more accurate value of 5730 yrs?
Figure 3b legend. Use a label that's more descriptive than "corr" for the corrected data; perhaps just say "corrected results"
The definition of the conventional $^{14}$C age is calculated from -8033 * ln (F$^{14}$C) defined by Stuiver and Polach (1977). This equation is based on the very original Libby half-life of 5568 years. This value was revised in the early 1960s to 5,730 ± 40 years, which meant that many calculated dates in papers published prior to this were incorrect. For consistency with these early papers, it was agreed at the 1962 Radiocarbon Conference in Cambridge (UK) to use the "Libby half-life" of 5568 years. Radiocarbon ages are thus still calculated using this half-life,

and are known as "Conventional Radiocarbon Age". Since the calibration curve (IntCal) also reports past atmospheric $^{14}$C concentration using this conventional age, any conventional ages calibrated against the IntCal curve will produce a correct calibrated age.

Figure 3b legend will be modified to "w/o carbonate contribution" indicated as without carbonate contribution.

Line 278. "As described in Section 3.2, no significant difference between F14C of DOC and WIOC was observed for the ice samples from Colle Gnifetti, Belukha and SLNS (Figure 3)." This is incorrect. Based on the figure, the differences seem significant at the 1-sigma level for several samples, and at the 2-sigma level for at least 1 sample.

Our data set, with the corresponding uncertainties does not allow us to conclude that there is a significant difference between F$^{14}$C of DOC and WIOC particularly for the data set from these three sites. We applied a Mann Whitney u-test (U=79.5, n1=n2=14, p=0.41>0.05) indicating the F$^{14}$C (DOC) and F$^{14}$C (WIOC) to be not significantly different. Easiest to see that this is true and that there is no statistical evidence of a difference is the fact that the 95% confidence interval in Figure 3 includes the 1:1 line. Of course, individual data points are expected to lie outside the 1 or 2 sigma level of the Gaussian distribution of the entire data set. This basically is the definition of these levels. Around 3 out of 10 individual data points are expected to be outside the one sigma range (68%) and around 1 out of 10 outside the 2 sigma range (95%). Therefore, the observation made by the reviewer is certainly correct but also exactly what is expected.

Anyhow, our formulation might not have been entirely clear and we will change to:

"As described in Section 3.2, there is no statistical evidence of a significant difference between F$^{14}$C of DOC and WIOC for the data-set from the Colle Gnifetti, Belukha and SLNS ice samples (Figure 3)."

Table 5 / Figure 4 and associated text. I think that the discussion of the limitations of this comparison to the previous age estimates should be expanded to provide some more detail / caveats associated with the comparison. For example, how close were the CG and Chongce cores to each other / do we even expect the basal ice to be of similar age? Is the comparison at Belukha still meaningful given the different core locations?

The two cores from CG were collected in 16 m distance from each other, the ones from Chongce at less than 2km distance from each other on this extended ice cap. Therefore we expect similar ages (for CG the age presented is not from basal ice). For Belukha we would also expect a similar glacier history, since both sites are only about 2 km apart and at the same elevation. We will add this information.

While I think that the authors' conclusion that there is no evidence that in situ 14C is affecting the 14C-DOC measurements is likely correct, the authors should do a better job of supporting this conclusion. For example, how can you be certain that the higher 14C-DOC as compared to 14C-WIOC in most samples is not due to in situ 14C? The largest offsets are observed at Chongce, which has the highest altitude and therefore should in principle have the highest in situ 14C production rates.

Thank you for this comment. We excluded an effect from $^{14}$C in-situ production in the initial version since no obvious super modern values were measured, in contrast to the findings of May (2009). However, thanks to the comments of both reviewers we realized, that this observation alone is not sufficient for such a conclusion.

We followed the suggestions of the reviewer and will add a section in the revised manuscript to more carefully estimate the potential of $^{14}$C in-situ production on DO$^{14}$C dating for each site.

For the production rate $P_o$ we used the literature values for different altitudes from Lal et al., 1987 in combination with the estimates for the latitudinal dependence of $P_o$ from Lal 1992. The annual accumulation rates for the new cores from CG, Belukha, and Chongce are not available at this point. Therefore, the according values were approximated based on previous studies for these sites. For CG from Jenk et al., 2009, for Belukha from Henderson et al., 2006, and for Chongce from Hou et al., 2018 for core3 (Table S1 and Table S2, see below). All these cores were drilled closeby of the new sites (see response below) and although some variation cannot be excluded, the potential difference is assumed to be relatively small with a negligible effect for the calculations here. For the SLNS core the annual accumulation rate has not been determined yet. Instead we estimated the annual accumulation rate (0.21 ±0.11 m w.e./yr ) by using a 2-dimensional glaciological flow model (2p model, Bolzan, 1985; Thompson et al., 1989) to fit the DO$^{14}$C dates. We find an estimated average offset of DOC-F$^{14}$C values due to in-situ production of 0.044 ±0.033. Generally, we find a good correlation between the observed F$^{14}$C (DOC)-F$^{14}$C (WIOC) offset and the calculated $^{14}$C in-situ contribution to DO$^{14}$C with the in-situ production explaining about 50% of the observed difference (R=0.82, see Figure below).

[Figure]

Further, as shown in this Figure, it is evident that the potential effect of in-situ production is strongest for the samples from Chongce. Based on the calculation, this is explained by the high altitude in combination with a low annual accumulation rate of this site (Table S1 and Table S2). For sites from lower altitude and/or characterized by higher accumulation rates, the contribution of $^{14}$C in-situ production to the DOC fraction is small and within the analytical uncertainty. In addition, the effect of in-situ production also depends on the carbon concentration, being lower the higher the concentration. In conclusion, under most conditions, in-situ production is not significant. Only for ice samples from extrem altitude, especially in combination with low accumulation rates, DO$^{14}$C dating results should be carefully interpreted. Under these conditions a potential contribution from $^{14}$C in-situ production cannot be excluded and could introduce an age bias exceeding the analytically derived age distribution.

Changes to manuscript:

New section about in-situ production (partly in supplement), new figures, adapted section about carbonates (discussion, also in abstract and conclusion – effect likely even smaller than estimated before), new section combining in-situ and carbonate effects.

Table S1. Characteristics of the study sites.

| Site | Coordinates Elevation | Location | Total Length (m) | Accumulation (m w.e.year$^{-1}$) | References |
|---|---|---|---|---|---|
| Colle Gnifetti | 45°55'45.7''N, 7°52'30.5''E 4450 m asl. | Western Alps Swiss-Italian border | 76 | 0.45[*] | Sigl et al., 2018 |
| Belukha | 49°48'27.7"N, 86°34'46.5"E 4055 m asl. | Altai Mountains Russia | 160 | 0.5[&] | Henderson et al., 2006 |
| SLNS | 38°42'19.35"N, 97°15'59.70"E 5337 m asl. | Shulenanshan Mountain China | 81 | 0.21[#] | Hou et al., submitted |
| Chongce core1 | 35°14'5.77"N, 81°7'15.34"E 6010 m asl. | Kunlun Mountain China | 134 | 0.14[+] | Hou et al., 2018 |

[*]Accumulation rate from previous publication for the core collected in 2003 from the same drilling sites with 16 m distance.

[&]Accumulation rate from core collected in 2001 at 90 m distance from the drilling site in 2018.

[#]Accumulation is estimated from 2p model using DO$^{14}$C dates.

[+]Accumulation rate is from Chongce core 3 that is located at the same plateau with less than 2 km distance.

Table S2. Estimated in-situ production contribution to the DOC fraction.

| Core section | Ice mass (g) | Carbon mass (µg) | Depth (m w.e.) | $P_o$ ($^{14}$C atom/g ice year ) | $^{14}$C in-situ (no. of atoms) | Change of F$^{14}$C in DOC fraction | DOC-WIOC F$^{14}$C Offset |
|---|---|---|---|---|---|---|---|
| CG110 | 171 | 18.9 | 55.8 | 328 | 1197 | 0.033±0.011 | 0.068±0.032 |
| CG111 | 207 | 25.5 | 56.3 | 328 | 1197 | 0.030±0.010 | 0.053±0.024 |
| CG112 | 248 | 23.6 | 56.7 | 328 | 1197 | 0.038±0.013 | 0.037±0.026 |
| CG113 | 246 | 29.5 | 57.0 | 328 | 1197 | 0.030±0.010 | 0.064±0.019 |
| Belukha412 | 172 | 28.5 | 142.7 | 286 | 921 | 0.017±0.006 | -0.052±0.026 |
| Belukha414 | 128 | 41.9 | 143.9 | 286 | 921 | 0.009±0.003 | 0.027±0.024 |
| Belukha415 | 102 | 23.7 | 144.5 | 286 | 921 | 0.012±0.004 | 0.043±0.043 |
| SLNS101 | 238 | 44 | 47.9 | 345 | 2666 | 0.044±0.011 | 0.070±0.050 |
| SLNS113 | 213 | 39.4 | 54.4 | 345 | 2656 | 0.044±0.011 | 0.089±0.050 |
| SLNS122 | 234 | 57.9 | 58.1 | 345 | 2651 | 0.033±0.008 | -0.034±0.047 |
| SLNS127 | 183 | 57.8 | 60.5 | 345 | 2647 | 0.026±0.006 | 0.029±0.047 |
| SLNS136 | 220 | 48.3 | 64.7 | 345 | 2641 | 0.037±0.009 | 0.135±0.047 |
| SLNS139 | 208 | 48.1 | 66.5 | 345 | 2638 | 0.035±0.008 | 0.058±0.046 |
| SLNS141-142 | 246 | 43.8 | 67.7 | 345 | 2636 | 0.045±0.011 | 0.061±0.047 |
| CC237 | 208 | 28.5 | 113.7 | 497 | 5371 | 0.120±0.040 | 0.275±0.054 |
| CC244 | 167 | 21.7 | 117.6 | 497 | 5353 | 0.126±0.042 | 0.161±0.051 |
| CC252 | 120 | 24.3 | 120.2 | 497 | 5341 | 0.080±0.027 | 0.231±0.051 |

References:

Bolzan, J.: Ice flow at the Dome C ice divide based on a deep temperature profile, J. Geophys. Res., 90, 8111–8124, 1985.

Gavin, D. G., Estimation of inbuilt age in radiocarbon ages of soil charcoal for fire history studies. Radiocarbon 43, 27-44, 2001.

Henderson, K., Laube, A., Gäggeler, H. W., Olivier, S., Papina, T., & Schwikowski, M. Temporal variations of accumulation and temperature during the past two centuries from Belukha ice core, Siberian Altai. Journal of Geophysical Research: Atmospheres, 111(D3), https://doi.org/10.1029/2005JD005830, 2006.

Hoffmann HM. Micro radiocarbon dating of the particulate organic carbon fraction in Alpine glacier ice: method refinement, critical evaluation and dating applications [PhD dissertation]. Combined Faculties for the Natural Sciences and for Mathematics of the Ruperto-Carola University of Heidelberg. (available at: http://archiv.ub.uniheidelberg.de/volltextserver/20712/ ), 2016.

Hoffmann, H., Preunkert, S., Legrand, M., Leinfelder, D., Bohleber, P., Friedrich, R., & Wagenbach, D. A New Sample Preparation System for Micro-14C Dating of Glacier Ice with a First Application to a High Alpine Ice Core from CG (Switzerland). Radiocarbon, 60(2), 517-533. doi:10.1017/RDC.2017.99, 2018.

Hou, S., Jenk, T. M., Zhang, W., Wang, C., Wu, S., Wang, Y., Pang, H. and Schwikowski, M. J. T. C.: Age ranges of the Tibetan ice cores with emphasis on the Chongce ice cores, western Kunlun Mountains, The Cryosphere, 12, 2341-2348, https://doi.org/10.5194/tc-12-2341-2018, 2018.

Hou, S., Zhang W., Fang L., Jenk T.M., Wu S., Pang H., Schwikowski M., Brief Communication: New evidence further constraining Tibetan ice core chronologies to the Holocene, Submitted to The Cryosphere.

Jenk, T. M., Ice core based reconstruction of past climate conditions and air pollution in the Alps using radiocarbon,Doctoral dissertation,University Bern, 2006.

Jenk, T. M., Szidat, S., Bolius, D., Sigl, M., Gaeggeler, H. W., Wacker, L., Ruff, M., Barbante, C., Boutron, C. F. and Schwikowski, M.: A novel radiocarbon dating technique applied to an ice core from the Alps indicating late Pleistocene ages, J Geophys. Res. Atmos., 114, https://doi.org/10.1029/2009JD011860, 2009.

Lal, D., K. Nishiizumi, and J. R. Arnold.: In situ cosmogenic 3H, 14C, and 10Be for determining the net accumulation and ablation rates of ice sheets, Journal of Geophysical Research: Solid Earth 92.B6 4947-4952, 1987.

Lal, Devendra: Cosmogenic in situ radiocarbon on the earth, Radiocarbon After Four Decades, Springer, New York, NY, 146-161, 1992.

Legrand, M., Preunkert, S., Jourdain, B., Guilhermet, J., Fain, X., Alekhina, I. and Petit, J. R.: Water-soluble organic carbon in snow and ice deposited at Alpine, Greenland, and Antarctic sites: a critical review of available data and their atmospheric relevance, Clim. Past Discuss., 9, 2357-2399, doi:10.5194/cpd-9-2357-2013, 2013.

May, B. L. Radiocarbon microanalysis on ice impurities for dating of Alpine glaciers, Ph.D. thesis, University of Heidelberg, Germany, 127pp., 2009.

May, B., D. Wagenbach, H. Hoffmann, M. Legrand, S. Preunkert, and P. Steier, Constraints on the major sources of dissolved organic carbon in Alpine ice cores from radiocarbon analysis over the bomb‐peak period, Journal of Geophysical Research: Atmospheres 118, 3319-3327,2013.

Minguillon MC, Perron N, Querol X, Szidat S, Fahrni SM, Alastuey A, et al. Fossil versus contemporary sources of fine elemental and organic carbonaceous particulate matter during the DAURE campaign in Northeast Spain, Atmos Chem Phys 11, 12067-12084, 2011.

Mohn, J., Szidat, S., Fellner, J., Rechberger, H., Quartier, R., Buchmann, B., & Emmenegger, L. Determination of biogenic and fossil CO2 emitted by waste incineration based on 14CO2 and mass balances. Bioresource Technology, 99(14), 6471-6479, 2008.

Sigl, M., Abram, N., Gabrieli, J., Jenk, T. M., Osmont, D., and Schwikowski, M., 19th century glacier retreat in the Alps preceded the emergence of industrial black carbon deposition on high-alpine glaciers, The Cryosphere,12,3311-3331, https://doi.org/10.5194/tc-12-3311-2018, 2018.

Stuiver, M., & Polach, H. A. Discussion reporting of 14 C data. Radiocarbon, 19(3), 355-363,1977.

Thompson, L. G., Mosley-Thompson, E., Davis, M., Bolzan, J., Dai, J., Klein, L., Yao, T., Wu, X., Xie, Z., and Gundestrup, N.: Holocene-late pleistocene climatic ice core records from Qinghai-Tibetan Plateau, Science, 246, 474–477, https://doi.org/10.1126/science.246.4929.474, 1989.

Zhang, Y., Y. Yao, X. Wang, Y. Liu, andS. Piao, Mapping spatialdistribution of forest age in China,Earthand Space Science,4, 108–116,doi:10.1002/2016EA000177, 2017.

---

## Author Response (AR1)

**Author's response to referee comments on: "Radiocarbon dating of alpine ice cores with the dissolved organic carbon (DOC) fraction" by Fang et al.,**

Correspondence to: Theo M. Jenk (theo.jenk@psi.ch)

We would like to thank the reviewers for their constructive comments that helped us to improve the accuracy of our evaluation of the potential of DOC for radiocarbon dating. Our responses to their comments are in blue.

**Anonymous Referee#1**

Received and published: 10 October 2020

The manuscript from Fang et al. investigates the possibility to use the dissolved organic carbon (DOC) fraction for 14C dating in high Alpine glacier ice. To do so the authors present an ice core sample set (17 ice core sections) taken from the deep parts of the high altitude Eurasian glaciers Colle Gnifetti, Belukha, Chongce, and Shule Nanshan, for which a direct 14C dating comparison between the water-insoluble organic carbon fraction (WIOC) and the DOC fraction was achieved for each sample. It should be noted that "direct comparison" means that each of the 17 ice core sections samples was cut lengthwise and WIOC as well a DOC 14C was measured on each ice core section, i.e. on exact the same depth interval of the ice core. Whereas the WIOC method is already well established, doubts were reported about suitability of the DOC fraction for 14C dating in an earlier study (May, 2009), what makes this study very challenging and important. After a short description of the deployed WIOC and DOC sample preparation methods, WIOC and DOC concentration as well as the radiocarbon results are presented and discussed. 3 of the 4 sites show almost identically (not significantly different within the error) 14C ages for the corresponding samples, with a slight but systematic offset towards higher F14C values for DOC compared to WIOC. For one site (Chongce) this offset is enhanced. Since this latter site contains a high influence of dust in the ice the observed F14C DOC-WIOC offset is discussed by testing the hypothesis of an incomplete removal of carbonate during the WIOC sample preparation using the Ca2+ concentration in the samples as tracer for calcium carbonate

The paper is well structured and written in almost all parts and addresses an important scientific question, which is in the scope of TC. The study presents an up to this point unique data set which is suitable and convincing for the discussed topic and most of the conclusions made, and for which I would like to felicitate the authors. The description of experiments and the presentation of the data as well as the discussion on the potential influence of incomplete removal of mineral dust on the WIOC sample preparation are except a few points (see my minor comments below) sufficiently complete and precise. Therefore I think the manuscript should be published after a few minor and one major revision were made.

Apart from the minor points which are listed below, my major concern is that the paper lacks a more detailed discussion about the potential influence of in-situ produced 14C on the DOC radiocarbon content in high altitude glacier ice. Present state of the art in literature is that this effect makes the use of DOC unsuitable for 14C dating, at least at low accumulation, high altitude mountain site as the Colle Gnifetti (denoted CG in the

following) (May 2009, Hoffmann 2016) from which samples are presented here. At present state of the manuscript, the authors state:

"The fact that none of the samples analyzed in this study (n=17) resulted in super modern F14C values (> 1) and the obtained significant correlation between the F14C of WIOC and DOC (Sect. 3.2) and the resulting calibrated 14C ages (Pearson r = 0.988, p < .01, n=14, Figure S1) represent strong evidence against the previously suggested 14C in-situ production in the DOC fraction (May, 2009)."

This argumentation could possibly be drawn referring to work from May (2009) only.Within this particular study 14C DOC measurements underlie relative high blank contributions and therefore a high uncertainty, and corresponding 14C POC data are likely influenced by altered soil and dust material incorporated in the ice due to high combustion temperatures. Thus the dataset of this study is very scattered and May(2009) could at that point only speculate on the existence of an in-situ 14C production on the DOC content in ice at CG.

However the work of Hoffmann (2016) proofed via neutron irradiation experiments that (i) the production of 14C in glacier ice and the incorporation into the DOC fraction is possible and (ii) gave a quantitative estimate of the DOC incorporated fraction of produced 14C in Alpine ice. Based on this, the study finally also details a way to calculate its influence on ice core samples from this site.

Since this work is not yet referenced in the present study, here a brief summary of what is outlined there:

In view that:

- 1) The production of 14C atoms within the ice matrix by spallation of oxygen within the water molecule, induced by cosmic radiation (cited references: Lal et al., 1987; van de Wal et al., 1994; Mazarik and Reedy, 1995) is a known process.
- Potential 14C production in organic compounds as CO and CO2, but also in CH4 (cited: Kemp et al., 2002, Petrenko et al., 2009, 2013), as well as the possibility to hydrogenate the CO molecule to higher organic species (cited: Woon, 2002) are already reported in literature

Hoffmann (2016) performed the irradiation experiment mentioned above to confirm or not what is proposed in literature. The experiment showed that 14C in-situ production in DOC is a real process and suggests that between 11-25 % of the initially produced 14C atoms entered into the DOC fraction of Alpine Glacier ice.

On the base of that, as outlined by Hoffman (2016),

- 1) the theoretically produced number of 14C atoms for mid latitude glacier site at an altitude of 4500 m asl can be estimated as a function of accumulation rate and depth (based on literature data), and
- 2) the relative amount of 14C, which entered in the DOC fraction of Alpine Glacier ice can be estimated quantitatively from the neutron irradiation experiment.

Since this study exists, and the in-situ production in the DOC fraction would result in enhanced F14C fractions, I think it is really worth and necessary to take this effect into account. It should be discussed in this manuscript as partial or at least potential cause of

the systematically observed DOC-WIOC difference, beside the hypothesis of the incomplete inorganic carbon removal within the WIOC sample preparation (which surely is also a good candidate for the observed offset in case mineral dust is present in the samples).

Having been curious myself on the order of magnitude this effect would have on the DOC 14C values measured in this study, I did a back-of-the-envelope calculation of the in-situ effect by applying the calculations of Hoffmann (2016) on the CG samples of this study. Accumulation rate and depth in water equivalent of CG15 are not given in the manuscript. Since however similar ages were found at similar depths in the cores CG15 and CG03 (see table 3 and section 4.3) the respective data from the CG03 (drilled in2003 almost directly at the saddle point of CG, Jenk et al., 2009) were used for the estimation. As the ice in the deeper part of the C15 core probably originates from upstream the drill site, i.e. from a position on the north flank of the CG, where the accumulation is lower (e.g. Licciulli et al., 2020), and since the accumulation rate is one of the driving factors of the magnitude of the in-situ production in the estimation, calculations for different accumulation rates were carried out. The mean of uppermost 30 years of CG3 (0.47 mwe/yr) was used, and additionally two values (0.25, and 0.12mwe/yr), which are in the order of magnitude of what is found upstream in the north flank of the CG. Since all four samples were taken from about the same depth, and had the same sample and carbon masses, mean values of (220g, 24ugC, and 56.75 mweq) were used in the calculation. As relative fraction of 14C, which entered in the DOC fraction 15% were assumed.

The estimation resulted in potential F14C offsets of 0.025, 0.047, and 0.096 for the assumed accumulation rates of 0.47 mwe/yr, 0.25, and 0.12 mwe/yr, respectively, which fits quite well with the observed offset within this study (0.055±0.014). Therefore, as stated above, a discussion of the in-situ production of 14C influencing theDOC 14C dating should not be neglected but done here. It would also significantly improve the scientific output of the manuscript. In addition, in view of the expected results, all existing studies on this topic would become conclusive and an important gap of knowledge in literature could be closed.

Thank you very much for this very valuable input and the details provided regarding a potential contribution of  ${}^{14}$ C in-situ production to DO ${}^{14}$ C. We excluded an effect from  ${}^{14}$ C in-situ production in the initial version since no obvious super modern values were measured, in contrast to the findings of May (2009). However, thanks to this comment we now are aware of the fact, that this observation alone is not sufficient for such a conclusion.

We followed the suggestions of the reviewer and will add a section in the revised manuscript to more carefully estimate the potential of 14C in-situ production on DO.14C dating for each site. For the production rate  $P_{o}$  we used the literature values for different altitudes from Lal et al., 1987 in combination with the estimates for the latitudinal dependence of  $P_{o}$  from Lal 1992. The annual accumulation rates for the new cores from CG, Belukha, and Chongce are not available at this point. Therefore, the according values were approximated based on previous studies for these sites. For CG from Jenk et al., 2009, for Belukha from Henderson et al., 2006, and for Chongce from Hou et al., 2018 for core3 (Table S1 and Table S2, see below). All these cores were drilled closeby of the new sites (see response below) and although some variation cannot be excluded, the potential difference is assumed to be relatively small with a negligible effect for the calculations here. For the SLNS core the annual accumulation rate has not been determined yet. Instead we estimated the annual accumulation rate (0.21 ±0.11 m w.e./yr ) by

using a 2-dimensional glaciological flow model (2p model, Bolzan, 1985; Thompson et al., 1989) to fit the DO14C dates. We find an estimated average offset of DOC-F14C values due to in-situ production of 0.044  $\pm$ 0.033. Generally, we find a good correlation between the observed F14C (DOC)-F14C (WIOC) offset and the calculated 14C in-situ contribution to DO14C with the in-situ production explaining about 50% of the observed difference (R=0.82, see Figure below).

Further, as shown in this Figure, it is evident that the potential effect of in-situ production is strongest for the samples from Chongce. Based on the calculation, this is explained by the high altitude in combination with a low annual accumulation rate of this site (Table S1 and Table S2). For sites from lower altitude and/or characterized by higher accumulation rates, the contribution of 14C in-situ production to the DOC fraction is small and within the analytical uncertainty. In addition, the effect of in-situ production also depends on the carbon concentration, being lower the higher the concentration. In conclusion, under most conditions, in-situ production is not significant. Only for ice samples from extreme altitude, especially in combination with low accumulation rates, DO14C dating results should be carefully interpreted. Under these conditions a potential contribution from 14C in-situ production cannot be excluded and could introduce an age bias exceeding the analytically derived age distribution.

Changes to manuscript:

New section (Sect. 4.2) about in-situ production (details of calculation in supplement), new figures, adapted section about carbonates (discussion, also in abstract and conclusion – effect likely even smaller than estimated before), new section combining in-situ and carbonate effects.

| Site | Coordinates | Location | Total        | Accumulation        | References |
|------|-------------|----------|--------------|---------------------|------------|
|      | Elevation   |          | Length       | $(m w.e.year^{-1})$ |            |
|      |             |          | ( m ) | •                   |            |

Table S1. Characteristics of the study sites.

| Colle
Gnifetti
(2015) | 45°55'45.7''N, 7°52'30.5''E
4450 m asl.  | Western Alps
Swiss-Italian
border | 76  | 0.45.* | Jenk et al.
2009; Sigl et                           |
|-----------------------------|---------------------------------------------|-----------------------------------------|-----|--------|--------------------------------------------------------|
| Belukha
(2018)           | 49°48'27.7"N, 86°34'46.5"E
4055 m asl.   | Altai
Mountains
Russia            | 160 | 0.5.*  | Henderson et
al., 2006;
Uglietti et al.,
2016 |
| SLNS
(2010)              | 38°42'19.35"N, 97°15'59.70"E
5337 m asl. | Shu Le Nan
Shan
Mountain
China | 81  | 0.21.# | Hou et al.,
submitted                               |
| Chongce
(2013,
Core1) | 35°14'5.77"N, 81°7'15.34"E
6010 m asl.   | Kunlun
Mountain
China             | 134 | 0.14.+ | Hou et al.,
2018                                    |

\*Previously reported value for a core collected from the same drilling site in 2003 (16 m distance).

&Previously reported value for a core collected from the same location in 2001 (90 m distance). #Estimate based on a glaciological flow model (2p model) and DO14C dated horizons. +Previously reported value for Chongce core 3, extracted less than 2 km away from the same

glacier plateau.

| Core section | Ice mass    | Carbon mass | Depth             | Po                                            | In-situ 14 C | In-situ F 14 C-DOC | Observed F 14 C | In-situ                      | In-situ       |
|---------------------|-------------|-------------|-------------------|-----------------------------------------------|-------------------------|-------------------------------|----------------------------|------------------------------|---------------|
|                     | (g ) | (µg)        | ( m w.e. ) | ( 14 C atom g -1 ice yr | (atoms)                 | offset                        | DOC-WIOC                   | corrected F 14 C- | corrected DOC |
|                     |             |             |                   | 1 )                                |                         |                               | offset                     | DOC                          | Cal age       |
| CC110               | 171         | 10.0        | FF 0       | 220                                           | 1107                    | 0.022+0.012                   | 0.00010.000                | 0.010+0.022                  | (cal DP)      |
| CGIIO               | 1/1         | 18.9        | 55.8              | 328                                           | 1197                    | 0.033±0.013                   | 0.068±0.032                | 0.910±0.033                  | 752±273       |
| CG111               | 207         | 25.5        | 56.3              | 328                                           | 1197                    | 0.030±0.012                   | 0.053±0.024                | 0.901±0.024                  | 1045±207      |
| CG112               | 248         | 23.6        | 56.7              | 328                                           | 1197                    | 0.038±0.015                   | 0.037±0.026                | 0.889±0.026                  | 1225±250      |
| CG113               | 246         | 29.5        | 57.0              | 328                                           | 1197                    | 0.030±0.012                   | 0.064±0.019                | 0.849±0.020                  | 1546±208      |
| Belukha412          | 172         | 28.5        | 142.7             | 286                                           | 921                     | 0.017±0.007                   | -0.052±0.026               | 0.315±0.025                  | 11271±902     |
| Belukha414          | 128         | 41.9        | 143.9             | 286                                           | 921                     | 0.009±0.003                   | 0.027±0.024                | 0.239±0.020                  | 14096±964     |
| Belukha415          | 102         | 23.7        | 144.5             | 286                                           | 921                     | 0.012±0.005                   | 0.043±0.043                | 0.144±0.041                  | 21571±4753    |
| SLNS101             | 238         | 44          | 47.9              | 345                                           | 2666                    | 0.044±0.017                   | 0.070±0.050                | 0.972±0.023                  | 587±187       |
| SLNS113             | 213         | 39.4        | 54.4              | 345                                           | 2656                    | 0.044±0.017                   | 0.089±0.050                | 0.942±0.023                  | 837±184       |
| SLNS122             | 234         | 57.9        | 58.1              | 345                                           | 2651                    | 0.033±0.013                   | -0.034±0.047               | 0.773±0.016                  | 2483±210      |
| SLNS127             | 183         | 57.8        | 60.5              | 345                                           | 2647                    | 0.026±0.010                   | 0.029±0.047                | 0.730±0.014                  | 2967±197      |
| SLNS136             | 220         | 48.3        | 64.7              | 345                                           | 2641                    | 0.037±0.014                   | 0.135±0.047                | 0.657±0.017                  | 4264±304      |
| SLNS139             | 208         | 48.1        | 66.5              | 345                                           | 2638                    | 0.035±0.014                   | 0.058±0.046                | 0.580±0.016                  | 5600±290      |
| SLNS141-142         | 246         | 43.8        | 67.7              | 345                                           | 2636                    | 0.045±0.018                   | 0.061±0.047                | 0.550±0.020                  | 6323±363      |
| CC237               | 208         | 28.5        | 113.7             | 497                                           | 5371                    | 0.120±0.046                   | 0.275±0.054                | 0.980±0.052                  | 1240±498      |
| CC244               | 167         | 21.7        | 117.6             | 497                                           | 5353                    | 0.126±0.049                   | 0.161±0.051                | 0.800±0.052                  | 3509±799      |
| CC252               | 120         | 24.3        | 120.2             | 497                                           | 5341                    | 0.080±0.031                   | 0.231±0.051                | 0.546±0.035                  | 7007±635      |

**Table 4** Estimate of the effect from in-situ  ${}^{14}$ C production on  $E^{14}$ C-DOC. For comparison, the measured  $E^{14}$ C offset between DOC and WIOC is also shown.

Minor comments:

Line 34-36: it would be good if you could give an idea of how much ice would be needed (inclusive the lost during decontamination) for an Antarctic sample (see also my comment on line 389).

In addition, be aware and mention that the potential in-situ effects will be much stronger in Polar Regions than in the high altitude sites in mid-latitudes, since the neutron flux and thus production rate is higher and accumulation rates are generally lower there.

In the abstract, we would like to point out the potential of pushing radiocarbon dating of ice forward even to more remote regions, where the carbon content in the ice is lower, when applying the DOC fraction for 14C dating. We will change the sentence mentioning remote regions. We think an estimation of in-situ in Polar Regions is outside the scope of this manuscript, but we agree that this is a topic to be looked at in future studies.

Line 48-50: ... Ice flow models, which are widely used to retrieve full depth age scales(e.g. Nye, 1963; Bolzan, 1985; Thompson et al., 2006), also fail in the deepest part of highalpine glaciers due to the complex bedrock geometry. ....

Please clarify or revise this sentence. To my knowledge the high model uncertainty in the deepest part of the glacier (which includes for me the deepest 5-10m above bedrock) its not only due to the bedrock geometry, but rather to the uncertainties in the assumptions needed to be made to constrain the model and which include beside the bedrock geometry also mass balance upstream, equation of temperature depended shear stress, steady state conditions.

This is correct. We will revise the sentence: "Ice flow models, ..., also fail in the deepest part of high-alpine glaciers due to the assumption of steady state conditions and the complexity of glacial flow and bedrock geometry limiting realistic modeling of strain rates."

Line 59-62: Samples of >10  $\mu$ g WIOC can be dated with reasonable uncertainty (10-20%), requiring less than 1 kg of ice from typical mid-latitude and low-latitude glaciers (Jenk et al., 2007; Jenk et al., 2009; Sigl et al., 2009; Uglietti et al., 2016). Please include also the study of Hoffmann et al., 2018, in which 14C dating on the WIOC franction was achived with an other sample preparation setup. Be also please more precise on the sample and carbon mass needed to achieve such an uncertainties of 10-20%. It seems that Hofmann et al. 2018 achieves this uncertainty with an ice mass <500g and a carbon mass of <10  $\mu$ gC. Also it would be good to mention whether the AMS or the sample preparation error dominates the uncertainty.

This statement is intended to give an idea, about what has been achieved so far. For this, we used an average estimate for a large range of samples from a variety of mid- to low-latitudes sites, varying in their age, and in their WIOC concentrations. For some sites, 200 g of ice is already sufficient (high concentrations) whereas other sites might require close to 1 kg. Whether it is the AMS or sample preparation error dominating the final dating uncertainty depends on the amount of 14C, and a general statement cannot be made. However, much more detailed information is accessible through the cited literature (Uglietti et al., 2016). We would like to note, that the estimate provided here is different in two ways from the numbers in Hoffmann et al. (2018) mentioned by the reviewer, The numbers of Hoffmann et al., (2018) are (1) defined for one specific site (Colle Gnifetti) only with uncertainties being indeed in the order of 10-20% for F14C values, but are (2) larger for the final calibrated ages. Anyhow, we agree with the reviewer that the reference of Hofmann et al., 2018 achieving at least similar precision for the CG site should be added, which we will do. We will rephrase the manuscript accordingly :

"Ice samples from mid- and low-latitude glaciers can now be dated with a reasonable uncertainty of 10-20%. Ice sample masses of 200-800 g are usually selected to aim for >10  $\mu$ g carbon for .14C analysis with accelerator mass spectrometry (AMS), whereby the respective mass depends on sample age and organic carbon concentrations (Jenk et al., 2007; Jenk et al., 2009; Sigl et al., 2009; Uglietti et al., 2016; Hoffmann et al., 2018)."

Line 79-81: In view of the analytical precision achievable with this method, the turnover time from atmospheric CO2 to deposited aerosol is negligible (Fang et al., in prep.). I am not sure if I got the meaning here.

Do you mean the analytical uncertainty, which results in an age error, which is much higher than the turn-over time?

Yes, the turn-over time from atmospheric CO2 to DOC is just a few years which is negligible compared to the width of the age distribution obtained from 14C dating due to the analytical uncertainty and calibration. We will rephrase the sentence in the revised version accordingly.

Line 93- 95: ... possible mechanisms of 14C in-situ formation in organic compounds seem far less likely and have not been investigated to date...

This sentence needs to be revised (see my major comment), since 14C in-situ formation in DOC of high Alpine glacier ice was investigated.

The sentence will be revised to "…possible mechanisms of 14C in-situ production followed by formation of organic compounds are far less understood and only few studies exist to date (Woon, 2002; Hoffmann 2016)."

Line 103-104: ....allowing 14C analysis on samples with DOC concentrations as low as  $25 \mu g/kg$  ....

I guess this assumption is made in view of the required carbon mass needed for 14C sample preparation and/or measurements. If true please mention that and change the sentence to something like:

The system can handle samples with volumes of up to  $\sim$ 350 mL. To achieve a minimal carbon mass required for 14C sample .....? A minimal DOC concentration of 25 µg/kg is needed.

The 25  $\mu$ g/kg DOC concentration is the detection limit of the DOC extraction setup (Fang et al., 2019). It was calculated based on 5-times of the average procedure blank (1.9)  $\mu$ gC. Considering the 350 ml maximum ice volume, the minimum carbon mass required for  $^{14}$ C analysis is thus 9.5  $\mu$ g. For samples with lower concentration, DOC could be extracted stepwise from more than one aliquot, but the corresponding blank needs to be determined.

The manuscript revised as: "The system can handle samples with volumes of up to ~350 mL. With this volume, samples with DOC concentrations as low as 25-30  $\mu$ g/kg can be analyzed, yielding the minimal carbon mass required for reliable.14C analysis (~10  $\mu$ g C)."

Line 116 – 133: It might be worth to summarize the meta data on the ice cores and samples listed here in a table (including geographic coordinates of the drill site, ice core lengths, accumulation rate at the drill site, sampled depths in this study, ... the mountain range and reference to study in which more meta data on the cores are given).

In any case at least the accumulation rate and the references to further meta data of the different cores should be added in the text.

The geographic coordinates of the drill sites, ice core lengths, depths of the samples analysed, the mountain range and the corresponding references in which more meta data on the cores are given, were provided in the manuscript (see section 2, Fig. 1 and Table 4 and

5). As for the accumulation rate, this information will now be included in a supplementary table (Table S1) as it is required for the calculation of  ${}^{14}$ C in-situ production. see comment above.

Line 185-187: ... and procedure blanks ( $1.26\pm0.59 \mu gC$  with F14C of  $0.69\pm0.15$  for WIOC samples and  $1.9\pm1.6 \mu gC$  with a F14C value of  $0.68\pm0.13$  for DOC samples)...

The way the WIOC and DOC procedure blanks were made and the frequency or number of blanks achieved during the analysis of this study should be given.

The information about the blank details will be given in the text now. "Procedural blanks were determined and continuously monitored by processing and analyzing frozen ultra-pure water (Sartorius, 18.2 M $\Omega$  cm, TOC< 5ppb) similar to natural ice samples. They were prepared every time when cutting ice and then processed/analyzed along with the samples at least twice a week. Procedural blanks are 1.3±0.6 µg C with an F14C of 0.69±0.15 (n=76) and 1.9±1.6 µg C with an F14C value of 0.68±0.13 (n=30) for WIOC and DOC, respectively."

Line 254 – 259: The fact that.... to ... (May, 2009).

In view of my request to discuss the potential bias due to 14C in-situ production by calculating its effect, these lines should be deleted.

This sentence has been deleted and new section 4.2 added.

Line 266 – 269: For DOC concentrations observed in this study, an initial ice mass of about 250 g was required, with about 20-30 % of the ice being removed during the decontamination processes inside the DOC set-up, yielding  $\sim$ 200 g of ice available for final analysis.

This sentence should be moved to Section 2 in the paragraph, which starts in line 156. We prefer to keep this sentence where it is. The reason is that it is a result of this study, providing a number of how much ice is needed, and what steps are required to yield  $DO^{14}C$  dating results within the precision and accuracy described.

Line 271:

Please specify here that the reduction of the sample mass in DOC refers to the WIOC method used at the PSI.

It was specified as "reduced by more than factor of two for required carbon mass". For these four sites, the required ice mass for  $WIO^{14}C$  is about 400-600 g depending on the concentration, but for  $DO^{14}C$  it is 100-300 g. Because this is mainly a result of the difference in concentrations of DOC compared to WIOC, we assume this factor of two to be a valid approximation independent of the set-up used.

**Line 276: Please add a section (4.2 or 4.3) on the "Potential contribution of 14C insituproduction to 14C of DOC" (see major comment)**

We will add one section to discuss the estimated  ${}^{14}C$  in-situ production offset as we mentioned in the response to the major comment (4.2 Potential contribution of  ${}^{14}C$  in situ production to DO ${}^{14}C$ ). For the calculation, details will also be added to supplementary.

Line 281-182: please change to something similar to:

... upper parts of the Chongce Cores 2 and 4, less than 2 and  $\sim$ 6 km away from Core 1, (measured with the same analytic device as used here), ...

It is clear in the method part that both are the same methodology and instrumentation as used.

Line 326-329: ... For final calibration of 14C ages, most of those earlier studiestook advantage of the assumption of sequential deposition in the archive, which seems very reasonable considering the deposition of annual snow layers on top of each other on the glacier surface.....

Please be more prudent here and revise this sentence since several studies emphasized that a sequential deposition in the archive of high Alpine glaciers is not evident (a least in the case for CG, see Jenk et al., 2009, Hoffmann et al., 2018, Bohleber, 2019). E.g. Bohleber 2019 wrote:

"... as already noted by Jenk et al. (2009), the finding of a continuous agedepth relation in the deep core parts is not a priori to be expected (e.g., as strong shear could potentially decouple the deformation of the basal ice frozen to bed from its adjacent top layer, which would be reflected in a hiatus in the age-depth relation). In fact, the 14C profile obtained by Hoffmann et al. (2018) for a core located on CG's north-facing slope (with significant bedrock inclination, cf. the saddle location of the core investigated by Jenk et al., 2009) revealed a localized discontinuity in 14C ages..."

Therefore I propose to argument like that:

- 1) Despite the fact that a sequential deposition in the archive is not evident in the deepest layers ... (references...)
- 2) but in view that in case of relatively large analytical uncertainties compared to the age difference of the samples, the sequential deposition model can moderately constrain the probability distribution of the calibrated age ....
  => The sequence model was used but results were compared using the conventional calibration approach. ...

Thank you for this comment. We agree with the reviewer that in individual cases, e.g. if there is indication in the data suggesting a hiatus or age inversion, the assumption of sequential deposition may not be valid. In such cases, this assumption needs to be discussed individually. This discussion however, exceeds the scope of the study here. Here, the idea was simply to treat the data precisely the same way as in the previous studies, to allow a direct comparsion with these previously published results. For the data in this study, there is no evidence that this assumption is invalid, in fact, as stated, the application of the sequence model has no effect on the final calibrated 14C ages (see table 4). Nevertheless, we did use the sequence results (see Table 3 and 5). In any case, to account for the point made by the reviewer we will change the sentence in question to:

" For final calibration of 14C ages, most of those earlier studies took advantage of the assumption of sequential deposition in the archive, i.e. a continuous, undisturbed and preserved sequential deposition of annual snow layers on the glacier surface. Particularly in case of relatively large analytical uncertainties compared to the age difference of the samples, the sequential deposition model can moderately constrain the probability distribution of the calibrated age range in each sample of the dataset. For consistency we applied the same calibration approach here by using the in-built OxCal sequence model (Ramsey, 2008). While the underlying assumption may not generally be valid for all sites, and individually needs to be carefully assessed, we find no difference in the calibrated ages using the sequence model and the ages from the conventional calibration approach for all DO14C data presented in this study (Table 3). Note, that no correction for a potential in-situ 14C bias was applied to the DO14C data used here (Section 4.2)."

Line 323: 4.3 DO14C ages in the context of published chronologies

In view of what is discussed in this paragraph I recommend to change the title to: D014C ages in the context of published near bedrock ice ages

The age of CG is not a bedrock age. Therefore we prefer to keep the more general subtitle.

Line 351 – 356, Table 5 and Figure 4:

1) to be complete for the CG site, please add also near bedrock ice age data obtained by Hoffmann et al., 2018 on an CG ice core (KCC) located on the north facing slope of theglacier, to the compilation of near bedrock ice ages. In the latter study the age difference of near bedrock ice between CG03 and the KCC is discussed, and might worth to bementioned that here.

We did not intend to compile Colle Gnifetti bedrock ages, but to compare our new results to previous results from the same site if such data is available. The KCC site is located on a different flow line with much lower ice thickness and results are not straightforward to compare. This is particularly the case very close to the bedrock, which is extensively discussed in Hoffmann et al., 2018. Because our aim here is to compare the new DOC dating results as directly as possible with dates from the validated WIOC method, and not to discuss the glacier flow or chronology of a specific site we think adding the data suggested by the reviewer would rather be confusing to the reader.

2) As already mentioned, the comparison of absolute depths between CG03 and CG15leads to assume that both ice cores were drilled at the same location of CG. If true addthis information in line 116

Yes, it is the same location. We will add the information under line 116 as: "A 76 m long core was retrieved from the glacier saddle in September 2015 at an altitude of 4450 m asl. (45°55'45.7''N, 7°52'30.5''E; Sigl et al., 2018) at 16 m distance from the location of the previously published CG03 core."

Line 389 ... This new dating method opens up new fields for radiocarbon dating of ice for example from remote or Polar Regions, where concentrations of organic impurities in the ice are particularly low ....

To illustrate this statement, please give an estimation of how much ice (in g or kg inclusive the ice mass which is needed for decontamination) would be necessary to achieve a 14C dating on an ice sample. Typical DOC concentrations from Antarctic ice with an for 14C dating accessible age ( < 10 ppb) are given e.g. in Legrand et al., 2013.

In addition as already stated in my comment to line 34-36, you should mention the potential influence of the 14C in-situ production which is expected to be enhanced compared to high altitude sites in mid latitudes, and will thus result in an enhanced age uncertainty.

We will rephrase the sentence to "more remote regions"

Literature cited in review:

Bohleber, Pascal. (2019). Alpine Ice Cores as Climate and Environmental Archives. 10.1093/acrefore/9780190228620.013.743.

Hoffmann HM. 2016. Micro radiocarbon dating of the particulate organic carbon fraction in Alpine glacier ice: method refinement, critical evaluation and dating applications [PhD dissertation]. Combined Faculties for the Natural Sciences and for

Mathematics of the Ruperto-Carola University of Heidelberg. (available at: http://archiv.ub.uniheidelberg.de/volltextserver/20712/)

Hoffmann, H., Preunkert, S., Legrand, M., Leinfelder, D., Bohleber, P., Friedrich, R., & Wagenbach, D. (2018). A New Sample Preparation System for Micro-14C Dating of Glacier Ice with a First Application to a High Alpine Ice Core from CG (Switzerland). Radiocarbon, 60(2), 517-533. doi:10.1017/RDC.2017.99

Jenk, T. M., Szidat, S., Bolius, D., Sigl, M., Gaeggeler, H. W., Wacker, L., Ruff, M., Barbante, C., Boutron, C. F. and Schwikowski, M.: A novel radiocarbon dating technique applied to an ice core from the Alps indicating late Pleistocene ages, J Geophys. Res. Atmos., 114, 457 https://doi.org/10.1029/2009JD011860, 2009.

Legrand, M. ; Preunkert, S. ; Jourdain, B. ; Guilhermet, J. ; Fain, X. ; Alekhina, I. A. ; Petit, J. R.: Water-soluble organic carbon in snow and ice deposited at Alpine, Greenland, and Antarctic sites: a critical review of available data and their atmospheric relevance. In: Climate of the Past 9 (2013), pp. 2195–2211

Licciulli, C., Bohleber, P., Lier, J., Gagliardini, O., Hoelzle, M. and Eisen, O.: A full Stokes ice-flow model to assist the interpretation of millennial-scale ice cores at the high-Alpine drilling site Colle Gnifetti, Swiss/Italian Alps, J. Glaciol., 66, 35-48, https://doi.org/10.1017/jog.2019.82, 2020.

May B. 2009. Radiocarbon microanalysis on ice impurities for dating of Alpine glaciers [PhD dissertation]. Institut für Umweltphysik, Heidelberg University.

Kemp, W.J.M. & Alderliesten, C. & van der Borg, Klaas & Jong, A.F.M. & Lamers, Robert-Jan & Oerlemans, J. & Thomassen, M. & Wal, R.S.W. (2002). In situ produced 14C by cosmic ray muons in ablating Antarctic ice. Tellus, Series B: Chemical and Physical Meteorology. 54. 10.1034/j.1600-0889.2002.00274.x.

Lal, D. ; Jull, A.J.T ; Donahue, D. J. ; Burtner, D. ; Nishizumi, K.: In situ cosmogenic 3H, 14C and 10Be for determining th net accumulation and ablation rates of ice sheets. In: Journal of Geophysical Research, Solid earth 92 (1987), Nr. B6, pp. 4947–4952.

Mazarik, J. ; Reedy, R. C.: Terrestrial cosmogenic-nuclide pro- duction systematics calculated from numerical simulations. In: Earth and Planetary Science Letters 136 (1995), pp. 381–395

Petrenko, V. V. ; Severinghaus, J. P. ; Smith, A. M. ; Riedel, K. ; Baggenstos, D. ; Harth, C. ; Orsi, A. ; Hua, Q. ; Franz, P. ; Takeshita, Y. ; Brailsford, G. ; Weiss, R. F. ; Buizert, C. ; Dickson, A. ; Schaefer, H.: High-precision 14C measurements demonstrate production of insitu cosmogenic 14CH4 and rapid loss of insitu cosmogenic 14CO in shallow

Greenland firn. In: Earth and Planetary Science Letters 365 (2013), pp. 190–197

Petrenko, V. V. ; Smith, A. M. ; Brook, E. J. ; Lowe, D. ; Riedel, K. ; Brailsford, G. ; Hua, Q. ; Schaefer, H. ; Reeh, N. ; Weiss, R. F. ; Etheridge, D. ; Severinghaus, J. P.: 14CH4 Measurements in Greenland Ice: Investigating Last Glacial Termination CH4 Sources. In: Science 324 (2009)

Wal, R. S. W. van de ; Roijen, J. J. van ; Raynaud, D. ; Borg, K. van der ; Jong, A. F. M. de ; Oerlemans, J. ; Lipenkov, V. Y. ; Huybrechts, P.: From 14C/12C measurements towards radiocarbon dating of ice. In: Tellus 46B (1994), pp. 94–102

Woon, D. E.: Modeling gas-grain chemistry with quantum chemical cluster calculations. I. heterogeneous hydrogenation of CO and H2CO on icy grain mantles. In: The Astrophysical Journal 569 (2002), pp. 541–548

**References:**

Bolzan, J.: Ice flow at the Dome C ice divide based on a deep temperature profile, J. Geophys. Res., 90, 8111–8124, 1985.

Henderson, K., Laube, A., Gäggeler, H. W., Olivier, S., Papina, T., & Schwikowski, M. Temporal variations of accumulation and temperature during the past two centuries from Belukha ice core, Siberian Altai. Journal of Geophysical Research: Atmospheres, 111(D3), https://doi.org/10.1029/2005JD005830, 2006.

Hoffmann HM. Micro radiocarbon dating of the particulate organic carbon fraction in Alpine glacier ice: method refinement, critical evaluation and dating applications,PhD dissertation, Ruperto-Carola University of Heidelberg, http://archiv.ub.uniheidelberg.de/volltextserver/20712/, 2016.

Hoffmann, H., Preunkert, S., Legrand, M., Leinfelder, D., Bohleber, P., Friedrich, R., & Wagenbach, D. A New Sample Preparation System for Micro-14C Dating of Glacier Ice with a First Application to a High Alpine Ice Core from CG (Switzerland). Radiocarbon, 60(2), 517-533. doi:10.1017/RDC.2017.99, 2018.

Hou, S., Jenk, T. M., Zhang, W., Wang, C., Wu, S., Wang, Y., Pang, H. and Schwikowski, M. J. T. C.: Age ranges of the Tibetan ice cores with emphasis on the Chongce ice cores, western Kunlun Mountains, The Cryosphere, 12, 2341-2348, https://doi.org/10.5194/tc-12-2341-2018, 2018.

Hou, S., Zhang W., Fang L., Jenk T.M., Wu S., Pang H., Schwikowski M., Brief Communication: New evidence further constraining Tibetan ice core chronologies to the Holocene, Submitted to The Cryosphere.

Jenk, T. M., Szidat, S., Bolius, D., Sigl, M., Gaeggeler, H. W., Wacker, L., Ruff, M., Barbante, C., Boutron, C. F. and Schwikowski, M.: A novel radiocarbon dating technique applied to an ice core from the Alps indicating late Pleistocene ages, J Geophys. Res. Atmos., 114, https://doi.org/10.1029/2009JD011860, 2009.

Lal, D., K. Nishiizumi, and J. R. Arnold.: In situ cosmogenic 3H, 14C, and 10Be for determining the net accumulation and ablation rates of ice sheets, *Journal of Geophysical Research: Solid Earth* 92.B6 4947-4952, 1987.

Lal, Devendra: Cosmogenic in situ radiocarbon on the earth, Radiocarbon After Four Decades, Springer, New York, NY, 146-161, 1992.

Legrand, M., Preunkert, S., Jourdain, B., Guilhermet, J., Fain, X., Alekhina, I. and Petit, J. R.: Watersoluble organic carbon in snow and ice deposited at Alpine, Greenland, and Antarctic sites: a critical review of available data and their atmospheric relevance, Clim. Past Discuss., 9, 2357-2399, doi:10.5194/cpd-9-2357-2013, 2013.

May, B. L. Radiocarbon microanalysis on ice impurities for dating of Alpine glaciers, Ph.D. thesis, University of Heidelberg, Germany, 127pp., 2009.

Sigl, M., Abram, N., Gabrieli, J., Jenk, T. M., Osmont, D., and Schwikowski, M., 19th century glacier retreat in the Alps preceded the emergence of industrial black carbon deposition on high-alpine glaciers, The Cryosphere, 12, 3311-3331, https://doi.org/10.5194/tc-12-3311-2018, 2018.

Thompson, L. G., Mosley-Thompson, E., Davis, M., Bolzan, J., Dai, J., Klein, L., Yao, T., Wu, X., Xie, Z., and Gundestrup, N.: Holocene-late pleistocene climatic ice core records from Qinghai-Tibetan Plateau, Science, 246, 474–477, https://doi.org/10.1126/science.246.4929.474, 1989.

**Author's response to referee comments on: "Radiocarbon dating of alpine ice cores with the dissolved organic carbon (DOC) fraction" by Fang et al.,**

Correspondence to: Theo M. Jenk (theo.jenk@psi.ch)

We would like to thank the reviewers for their constructive comments that helped us to improve the accuracy of our evaluation of the potential of DOC for radiocarbon dating. Our responses to their comments are in blue.

**Anonymous Referee #2**

Received and published: 10 October 2020

This manuscript presents the first results of a technique that utilizes 14C of DOC to date alpine ice cores. The basal sections of alpine ice cores are difficult to date because of high degree of ice thinning (making layer counting impossible) and complex ice flow. The approach used in this study is very analytically challenging, and in my opinion the method has been carefully developed and tested. The analytical precision on the F14C values is impressive considering the small sample sizes. Considering that this approach requires smaller ice samples (\_250 g) than the earlier approach developed by the same group that uses insoluble organic carbon, this now seems like the most promising technique for dating alpine basal ice. Overall, I think that this is an exciting study that is in principle well suited for The Cryosphere. However, I think the study and manuscript also have some weaknesses that should be addressed.

**Major Comments:**

One of the major goals of this study / manuscript is validation of the 14C-DOC technique. In my opinion the manuscript doesn't fully achieve this. The main approach for this evaluation is comparison with the WIOC-14C results. But those results seem to be affected (to varying degrees) by 14C interference from carbonate dust in the samples. Are the authors able to measure a few samples from a layer-counted Greenland ice core, for example, to provide a more robust validation? I realize that this may be difficult, both because of lower DOC concentrations and ice availability, but perhaps a core from a coastal ice cap such as Renland could be a good target?

As we discuss in the manuscript, the carbonate effect on WIOC is small and within the range of the analytical uncertainty (Figure S2). The ages of ice obtained with WIOC were validated before by comparison with independently dated ice (by annual layer counting or conventional 14C dating of microfossils contained in the ice) and this was published and we cite that (Uglietti et al., 2016). We therefore don't see the necessity to measure further samples from Greenland to validate the WIOC dating method used as benchmarker in this study.

I think it would be valuable to provide a more complete analysis of the overall dating uncertainties. If 14C-WIOC is the benchmark measurement that is being used for validation of 14C-DOC, then the uncertainties in the 14C-DOC ages need to fully reflect the uncertainties associated with us 14C-WIOC (see more on this below). Alternatively, if the authors consider 14C-DOC to be an inherently superior approach (as compared to 14C-WIOC), then a more clear argument needs to be made for this. I think the uncertainties associated with the correction for carbonate dust (for WIOC-14C) need to be more thoroughly considered. The authors provide some helpful discussion of this in the supplement (starting on line 33), but I'm not convinced that the uncertainties are fully accounted for. For example, it seems to me that F14Ccarb could in principle range from 0 to 1 depending on the source of the carbonate. One could imagine a situation with seasonally-drying lakes in arid regions, for example, where the carbonate dust at the surface would be close to modern in its 14C signature. The C /Ca ratio in dust derived from dolomite would be twice as large as what is being used in Supplement equation 2. The effect of these additional uncertainties may be visible in figure 3b – while the

correction makes the Chongce samples look more reasonable, two of the Belukha samples now fall off the trend.

Regarding the main issues raised by both reviewers, we do understand the concerns and would like to thank for the careful evaluation of the manuscript. While details can certainly be discussed, we however are a bit surprised that the reviewer here asks for an even more detailed analysis of the overall dating uncertainties. This considering the fact, that we discuss discrepancies, which are barely statistically significant (below the analytical detection limit for 4 out of 3 sites). Since the method of 14C-WIOC dating has been validated previously (see Uglietti et al., 2016) we also think that it is justified to use this as a benchmark. Anyway, in the revised version of the manuscript, we included the valuable suggestions of the two reviewers to even further improve this in-depth discussion (Sect.4.2 and 4.3). Consequently, with the new consideration of potential in-situ contribution to  $DO^{14}C$ , this fraction will no longer be considered as the superior approach. Instead, for both fractions, we are confident to be able to provide more precise and accurate guidelines about potential limitations in the accuracy for both approaches.

In the new calculation about in-situ production, we find about 50% of the offset between  $F^{14}C$ DOC-WIOC can be explained by in-situ production, see related comment. Although numbers of carbonates contribution to WIOC will thus change, the related modeling approach will still be part of the manuscript. First, we would like to stress, that this modeling results should not be viewed as a mean for correction of WIOC F14C results. Instead, the aim was to test the hypothesis that a less than 100% efficient carbonate removal procedure could potentially explain the observed offset between  $F^{14}C$  of WIOC and DOC with the required level of efficiency being plausible (high and only slightly less than 100%). As we stated, the future aim would be to improve the carbonate removal process, not to correct WIO14C for a potential carbonate bias. Under this aspect, we agree with the reviewer, that the selection of the parameter space for F14Ccarb and the carbonate-to-calcium ratio is critical. This is what we already stated in the supplement. Our opinion is that therein (lines 33-42 as pointed out by the reviewer), the range of possible F14C values for carbonates or carbonate-to-calcium ratios was already reasonably explored. We discussed the robustness of the modeling results, provided an idea of the associated uncertainty by exploring the parameter space. However, not intending to apply a correction for (a potential) carbonate bias, we thereby did not focus to precisely quantify uncertainties but considered an evaluation of those to be sufficient for determining if the model results are robust in terms of the estimated removal efficiency required for explaining the observed offset. For explanation in this response only, we will summarize below the reasoning behind our approach and point out what was already addressed in the supplement:

(1) The carbonate removal efficiency by the acidification step very likely depends on the source and transport of the mineral dust (geological form, particle size) affecting the solubility. Thus, tuning the model for a common carbonate removal efficiency ( $x_{eff}$ ) instead of allowing this parameter to vary for individual sites will likely will not yield the best possible approximation between the observations and the modeled data as reflected in Figure 3b. In the Supplement, line 38-45, we thus discussed results with a model set-up allowing the carbonate removal efficiency to be different for the individual sites (within +- 4% for all sites). This showes, that the estimated value for  $x_{eff}$  may not be a precise, best value for each site but a robust average estimation.

(2) As pointed out by the reviewer, the abundance of carbonates in their different geological forms is likely different for each site (similar to point (1) dependent on the source region) and thus the value for the C/Ca ratio is likely not a fixed single value. However, we do not know what value would be most appropriate for each of the sites. Allowing this parameter

to be free in the model, i.e. allowing it to tune to a "best" value for each individual site would thus be speculative and over-tuning of the model. Anyhow, please note, that in the Supplement we did some evaluation of changes to this parameter by using a different value for this ratio (0.8 instead of 0.5), which again provided an estimate about the robustness/uncertainty of the final result for  $x_{eff}$ .

(3)  $F^{14}C_{carb}$  very likely differs dependent on the site (again relates to the source). Best,  $F^{14}C_{carb}$  would be selected individually for each sample, based on the difference in age of the contemporary atmosphere to the assumed age of the source carbonate at that time. But also here, the available literature is sparse and without speculation we cannot assume "precise" values. The value suggested by the reviewer ( $F^{14}C_{carb} = 1$ ) would certainly not be a reasonable value to assume, it reflects the year 1950 AD. We think his idea was that at the time when a certain snow/ice layer now at some depth in the glacier was on the surface, mineral dust (carbonates) with an  $F^{14}C$  being contemporary at that time were deposited as impurities in this layer. In this case the upper limit for  $F^{14}C_{carb}$  would be equal to the  $F^{14}C$  corresponding to the age of this layer. It is clear, that a close to contemporary  $F^{14}C$  would not introduce a bias and carbonates then could not explain the observed offset between DOC and WIOC. However, layers from "input from seasonally-drying lakes in arid regions" may occur as individual, special events but as such do not represent the norm (also for some sites, this can be excluded as a possibility entirely). Anyway, in the Supplement we already calculated using a different value for  $F^{14}C_{carb}$  (0.05).

To conclude, we are very much aware that if the intention would be to model the observed offset as closely as possible to come up with a correction, the model-set up and parameter space could be explored in even more detail (note, that in principle, by tuning each parameter for individual sample, we can reproduce the offset to nearly 100%). However, in our opinion this would only be justified if knowing with absolute certainty that a contribution from carbonate carbon to F14C of WIOC exists and one could start to quantify and investigate the individual parameter values in detail. Again, we would like to point out that we look at discrepancies very close to the detection limit (or even below). Here we can thus only provide evidence that such an effect exists, causing a systematic offset in the direction observed. As mentioned in the beginning, now being aware that in-situ 14C contribution to DOC cannot be completely ruled out, the potential "carbonate bias" becomes even smaller (i.e. the removal efficiency even higher) than what we estimated before.

**Minor Comments:**

Line 73. For water-soluble organics sourced from biomass burning, there may be an age offset due to the older ages of the burned material. While this is probably small compared to the measurement uncertainties, it would still be worth mentioning briefly. Similar comment for organics sourced from oceanic emissions (affected by ocean radiocarbon reservoir effect).

We agree with the reviewer that there is an in-built radiocarbon age from old carbon reservoirs to be assumed, actually both, for the DOC and WIOC fractions. We added one paragraph to the manuscript as suggested:

"For both WIOC and WSOC, carbon from biomass burning and oceanic organic matter can potentially introduce a reservoir effect (sources of aged carbon). The mixed age of trees in Swiss forests today is estimated to be slightly less than 40 years (Mohn et al., 2008). Back in time, prior to extensive human forest management, the mixed age of trees in Europe was likely older and the mean age of old-growth forest wood ranged from around 70 to 300 years depending on the region, i.e. the tree species present (Gavin, 2001, Zhang et al., 2017). Prior to the use of fossil fuels about 50% of WIOC is estimated to originate from biomass burning (Minguillon et al., 2011). For biogenic DOC, May et al. (2013) estimated a turnover-time of

around 3 to 5 years, corresponding to a 20% contribution from biomass burning. With a mean age of burned material (aged wood plus grass and bushes) of  $150\pm100$  years, this results in a potential in-built age from biomass burning for WIOC and DOC of  $75\pm50$  and  $30\pm20$  years, respectively. Such an in-built age is negligible considering the analytical uncertainty, which is similarly the case for a bias from oceanic sources, since concentrations of marine organic tracers are more than one order of magnitude lower than terrestrial tracers for the vast majority of glacier sites. This conclusion is supported by the fact that Uglietti et al. (2016) did not identify such a bias, when comparing WIO.14C ages with ages derived by independent methods."

Line 191. Why is the Libby half-life of 14C being used here instead of the more accurate value of 5730 yrs?

Figure 3b legend. Use a label that's more descriptive than "corr" for the corrected data; perhaps just say "corrected results"

The definition of the conventional  ${}^{14}$ C age is calculated from -8033 \* ln (F14C) defined by Stuiver and Polach (1977). This equation is based on the very original Libby half-life of 5568 years. This value was revised in the early 1960s to 5,730 ± 40 years, which meant that many calculated dates in papers published prior to this were incorrect. For consistency with these early papers, it was agreed at the 1962 Radiocarbon Conference in Cambridge (UK) to use the "Libby half-life" of 5568 years. Radiocarbon ages are thus still calculated using this half-life, and are known as "Conventional Radiocarbon Age". Since the calibration curve (IntCal) also reports past atmospheric. 14C concentration using this conventional age, any conventional ages calibrated against the IntCal curve will produce a correct calibrated age.

Figure 3 changed to new figure. The carbonates corrected results are shown in Figure 5 in the revised manuscript.

Line 278. "As described in Section 3.2, no significant difference between F14C of DOC and WIOC was observed for the ice samples from Colle Gnifetti, Belukha and SLNS (Figure 3)." This is incorrect. Based on the figure, the differences seem significant at the 1-sigma level for several samples, and at the 2-sigma level for at least 1 sample.

Our data set, with the corresponding uncertainties does not allow us to conclude that there is a significant difference between  $F^{14}C$  of DOC and WIOC particularly for the data set from these three sites. We applied a Mann Whitney U-test (U=79.5, n1=n2=14, p=0.41>0.05) indicating the  $F^{14}C$  (DOC) and  $F^{14}C$  (WIOC) to be not significantly different. Easiest to see that this is true and that there is no statistical evidence of a difference is the fact that the 95% confidence interval in Figure 3 includes the 1:1 line. Of course, individual data points are expected to lie outside the 1 or 2 sigma level of the Gaussian distribution of the entire data set. This basically is the definition of these levels. Around 3 out of 10 individual data points are expected to be outside the one sigma range (68%) and around 1 out of 10 outside the 2 sigma range (95%). Therefore, the observation made by the reviewer is certainly correct but also exactly what is expected.

Anyhow, our formulation might not have been entirely clear and we changed to the Sect. 3.2 : "For three of the sites (Colle Gnifetti, Belukha and SLNS), the corresponding DOC and WIOC fractions yielded comparable  $F^{14}C$  values with no statistical evidence for a significant difference (Mann-Whitney U-test, U=79.5, n=14, p=0.41>0.05). They scatter along the 1:1 ratio line, are significantly correlated (Pearson correlation coefficient r=0.986, p < .01, n=14) and both intercept (0.025 ± 0.034) and slope (1.034 ± 0.050) are not significantly different from 0 and 1, respectively (Figure 3)."

Table 5 / Figure 4 and associated text. I think that the discussion of the limitations of this comparison to the previous age estimates should be expanded to provide some more detail /

caveats associated with the comparison. For example, how close were the CG and Chongce cores to each other / do we even expect the basal ice to be of similar age? Is the comparison at Belukha still meaningful given the different core locations?

The two cores from CG were collected in 16 m distance from each other, the ones from Chongce at less than 2km distance from each other on this extended ice cap. Therefore we expect similar ages (for CG the age presented is not from basal ice). For Belukha we would also expect a similar glacier history, since both sites are only about 2 km apart and at the same elevation. We added this information in the Sect. 2 and Table S1.

While I think that the authors' conclusion that there is no evidence that in situ 14C is affecting the 14C-DOC measurements is likely correct, the authors should do a better job of supporting this conclusion. For example, how can you be certain that the higher 14C-DOC as compared to 14C-WIOC in most samples is not due to in situ 14C? The largest offsets are observed at Chongce, which has the highest altitude and therefore should in principle have the highest in situ 14C production rates.

Thank you for this comment. We excluded an effect from 14C in-situ production in the initial version since no obvious super modern values were measured, in contrast to the findings of May (2009). However, thanks to the comments of both reviewers we realized, that this observation alone is not sufficient for such a conclusion.

We followed the suggestions of the reviewer and will add a section in the revised manuscript to more carefully estimate the potential of 14C in-situ production on DO14C dating for each site. For the production rate Po we used the literature values for different altitudes from Lal et al., 1987 in combination with the estimates for the latitudinal dependence of Po from Lal 1992. The annual accumulation rates for the new cores from CG, Belukha, and Chongce are not available at this point. Therefore, the according values were approximated based on previous studies for these sites. For CG from Jenk et al., 2009, for Belukha from Henderson et al., 2006, and for Chongce from Hou et al., 2018 for core3 (Table S1 and Table S2, see below). All these cores were drilled closeby of the new sites (see response below) and although some variation cannot be excluded, the potential difference is assumed to be relatively small with a negligible effect for the calculations here. For the SLNS core the annual accumulation rate has not been determined yet. Instead we estimated the annual accumulation rate ( $0.21 \pm 0.11$  m w.e./yr) by using a 2-dimensional glaciological flow model (2p model, Bolzan, 1985; Thompson et al., 1989) to fit the DO14C dates. We find an estimated average offset of DOC- $F^{14}C$  values due to in-situ production of 0.044  $\pm$ 0.033. Generally, we find a good correlation between the observed F14C (DOC)-F14C (WIOC) offset and the calculated 14C in-situ contribution to DO14C with the in-situ production explaining about 50% of the observed difference (R=0.82, see Figure below).